**REPORT**

# Lift-out cryo-FIBSEM and cryo-ET reveal the ultrastructural landscape of extracellular matrix

Bettina Zens[1], Florian Fäßler[1], Jesse M. Hansen[1], Robert Hauschild[1], Julia Datler[1], Victor-Valentin Hodirnau[1], Vanessa Zheden[1], Jonna Alanko[1], Michael Sixt[1], and Florian K.M. Schur[1]

**The extracellular matrix (ECM) serves as a scaffold for cells and plays an essential role in regulating numerous cellular processes, including cell migration and proliferation. Due to limitations in specimen preparation for conventional room-temperature electron microscopy, we lack structural knowledge on how ECM components are secreted, remodeled, and interact with surrounding cells. We have developed a 3D-ECM platform compatible with sample thinning by cryo-focused ion beam milling, the lift-out extraction procedure, and cryo-electron tomography. Our workflow implements cell-derived matrices (CDMs) grown on EM grids, resulting in a versatile tool closely mimicking ECM environments. This allows us to visualize ECM for the first time in its hydrated, native context. Our data reveal an intricate network of extracellular fibers, their positioning relative to matrix-secreting cells, and previously unresolved structural entities. Our workflow and results add to the structural atlas of the ECM, providing novel insights into its secretion and assembly.**

## Introduction

The extracellular matrix (ECM) is an intricate three-dimensional assembly of macromolecules and signaling factors and acts as a physical scaffold for ECM-residing cells. It controls cellular activities, such as proliferation or migration, via its biochemical, biomechanical, and biophysical properties. These properties are tissue-specific depending on the origin of ECM-producing cells (Frantz et al., 2010; Theocharis et al., 2016). Aberrant ECM composition and remodeling contribute to disease progression, and alterations in the ECM are associated with aging, cancer metastasis, and fibrosis (Robins, 2007; Frantz et al., 2010; Iozzo and Gubbiotti, 2018).

The ECM consists mostly of two classes of macromolecules: fibrous proteins (FPs) and proteoglycans (PGs). FPs include different types of collagens (Col, with 28 types existing in humans) (Ricard-Blum, 2011), fibronectins (FN), fibrillins, or elastins. They assemble the ECM scaffold and present soluble growth factors to cells (Frantz et al., 2010; Theocharis et al., 2016; Alberts et al., 2022). PGs form complexes with FPs or glycosaminoglycans. Extensive interconnections between FPs and PGs are required for ECM fiber assembly and maintenance (Danielson et al., 1997; Corsi et al., 2002; Sottile and Hocking, 2002; Chen et al., 2020).

While we have a thorough understanding of the molecular inventory of the ECM (Naba et al., 2012; Byron et al., 2013; Taha and Naba, 2019), the structural landscape of ECM fibrils and their interactions remain uncharted. Current approaches for studying ECM structures are mostly limited to the visualization of chemically contrasted specimens, where the employed conventional room temperature (RT) electron microscopy (EM) techniques are destructive to the strongly hydrated ECM environment. Finer details and importantly the molecular assembly of ECM components into fibrils or other higher-order arrangements cannot easily be discerned from such fixed and dehydrated ECM preparations. Beyond certain types of D-spaced collagen assemblies with a defined repeat length of 67 nm (e.g., Col-I or II) (Smith, 1968; Bozec et al., 2007; Stylianou, 2022), current literature does not allow for an unambiguous consensus on the exact molecular assembly of many of the other ECM fibers, such as FN fibrils, fibrillin microfibrils, or Col-VI filaments. For these fibers, studies employing different EM or super-resolution fluorescence microscopy approaches reported different fiber dimensions with varying diameters or repeat lengths (Table S1). For example, Col-VI was suggested to form a unique fiber assembly among the collagen superfamily with a 105–112 nm beaded repeat (Furthmayr et al., 1983; Baldock et al., 2003; Knupp et al., 2006). In contrast, immunogold labeling cryo-scanning transmission electron tomography (cryo-STET) data measured a Col-VI repeat pattern of only 85 nm (Lansky et al., 2019). The challenge in reconciling these different observations is due to the use of varying experimental modalities, the fact that some ECM fibers are indeed variable in the sizes they can grow into, and that most measurements have been conducted under conditions not representative of physiological

[1]Institute of Science and Technology Austria (ISTA), Klosterneuburg, Austria.

Correspondence to Florian K.M. Schur: florian.schur@ist.ac.at.

ECM assembly. A case in point are collagen fibers that spontaneously assemble *in vitro*, but not *in vivo* where they require additional proteins (Kadler et al., 2008). Furthermore, FN, fibrillins, and elastins assemble deformable fibers (Glab and Wess, 2008; Klotzsch et al., 2009) and are able to adapt to the tension exerted by cells and tissue.

The intricate and complex interplay between ECM components can be best studied within a native environment, such as in the context of matrix-secreting cells. However, this has been exacerbated by technological limitations in cryo-electron tomography (cryo-ET), a method that enables visualizing specimens in 3D under virtually artifact-free conditions (Wagner et al., 2017). Specifically, the thickness of ECM specimens, extending into the tens of micrometer range, requires sample thinning steps, for example, cryo-focused ion beam scanning electron microscopy (FIBSEM). Cryo-FIBSEM has been applied to a variety of specimens, i.e., when using bulk milling on isolated adherent cells (Marko et al., 2007; Rigort et al., 2010) or even small organisms, such as *C. elegans* (Harapin et al., 2015). More recently, the lift-out technique has been introduced to obtain lamellae of samples that are otherwise incompatible with conventional bulk milling approaches (Mahamid et al., 2015; Wagner et al., 2017).

Another aggravating factor for the structural annotation of ECMs is the heterogeneity of tissue-derived ECM material. In contrast, cell-derived matrices (CDMs) are a versatile tool increasingly used to mimic and study fundamental aspects of native tissue ECM (Hakkinen et al., 2011; Petrie and Yamada, 2016; Kaukonen et al., 2017; Cukierman et al., 2001). To obtain CDMs, ECM-producing cells such as fibroblasts are cultured over several weeks to produce a 3D matrix that closely resembles the tissue these cells originate from (Franco-Barraza et al., 2016). Given their single-cell type origin, CDMs provide the advantage of high reproducibility, genetic tractability, and homogeneity. CDMs are used in fundamental research focusing on cell motility and cell proliferation, as well as tissue engineering and regenerative medicine (Hakkinen et al., 2011; Fitzpatrick and McDevitt, 2015; Jacquemet et al., 2015; Rubi-Sans et al., 2020; Rubí-Sans et al., 2021). The adaptability of CDMs to distinct research questions has also made them a routinely used tool in cancer research (Franco-Barraza et al., 2017; Malik et al., 2019; Padhi et al., 2020; Francescone et al., 2021; Jones et al., 2022). The use of cancer-associated fibroblasts (CAFs) for CDM generation results in a close mimic of the tumor-associated stroma (Amatangelo et al., 2005) and has aided in the identification of novel regulators of ECM alignment (Jones et al., 2022) as well as potential therapy targets (Francescone et al., 2021). Here, we have implemented CDMs as a cell culture-based 3D-ECM platform, which is compatible with cryo-lift-out FIB milling and cryo-ET. Our cryo-electron tomograms of CDM reveal an intricate network of ECM fibers in the context of matrix-secreting cells and provide further insights into the still open questions on the molecular assembly of ECM components into a functional 3D matrix.

## Results and discussion

### On-grid CDM growth and characterization

We adapted a previously published protocol (Kaukonen et al., 2017) to render CDMs produced by human telomerase immortalized foreskin fibroblasts (TIFFs) compatible with cryo-lift-out FIBSEM and cryo-ET experiments. Employing grid holders to facilitate long-term cell culture on EM-grids (Fäßler et al., 2020), we performed time course studies to determine the optimal culture conditions and growth time to generate fully formed CDMs (Fig. 1, A and B, see Materials and methods). Collagen secretion from cells could be already observed shortly after reaching cell confluency (designated as Day 0), and on Day 7 collagen fibers had formed. After 14 days, collagen assembly in CDMs reached an average height of 14.8 µm ($n = 7$; SD = ± 2.8 µm) when measured in confocal microscopy, which did not substantially increase when cultivated for longer time points. Mass spectrometry of Day 14 CDMs identified 110 ECM proteins, supporting that at this time point a full matrix assembly had formed (Table S2). Correspondingly, FN fibers were also highly enriched in our Day 14 CDMs when visualized via immunofluorescence microscopy, which also revealed a multilayer of cells embedded within ECM (Fig. 1 C). A detailed analysis of our CDM proteomics data revealed fibrillar ECM proteins making up >50% of the complete ECM protein fraction (Fig. 1 D). Col-I, Col-VI, and FN I were the most abundant, followed by other collagens and Fibrillin-1 (Fig. 1 E).

To further characterize our CDMs, we performed RT array tomography via scanning electron microscopy (SEM) of thin-sectioned CDMs (Fig. S1). Array tomography sacrifices nativity and resolution due to the involved chemical fixation and dehydration process, but allows large-field volumetric imaging with improved resolution in the Z-axis compared with confocal imaging. Segmentation of the 3D array tomography data revealed ~10 cells overlapping each other over a height of 15 µm, adopting a flat and extended morphology. Cells were embedded in ECM, as judged by the presence of what we assumed to be collagen fibers (Fig. S1, A–C). ECM and cells occupied 36% and 64%, respectively, of the total imaged volume. Cell and ECM fiber orientation showed a clear dependence on the Z-height of the CDM (Fig. S1 D), where fibers aligned with the long axis of the cell. Previous publications have reported the direct influence of cells, specifically fibroblasts, on the orientation of ECM fibers, which is mediated by cytoskeleton–ECM interactions via adhesion complexes (Geiger et al., 2001; Harris et al., 1981). Cells align ECM fibers such as FN and collagens by exerting forces through cell–ECM adhesion interactions (Piotrowski-Daspit et al., 2017). Altogether, these results confirmed that CDMs harvested on or later than Day 14 mimic ECM assemblies found in tissue (Fitzpatrick and McDevitt, 2015; Ahlfors and Billiar, 2007) and should allow visualization of ECM components in their native environment.

### Vitrification optimization and correlative imaging of CDMs

Due to their height and comparatively high free water content, CDMs exceed the vitrification potential of plunge freezing. Hence, we performed high-pressure freezing (HPF) of on-grid CDMs (Fig. S2 A). Vitrification status after HPF can only be evaluated in cryo-transmission electron microscopy (cryo-TEM) via observation of ice crystal reflections in thinned lamellae. Hence, upon vitrification, we first imaged fluorescently labeled CDMs via cryofluorescence light microscopy to judge CDM and

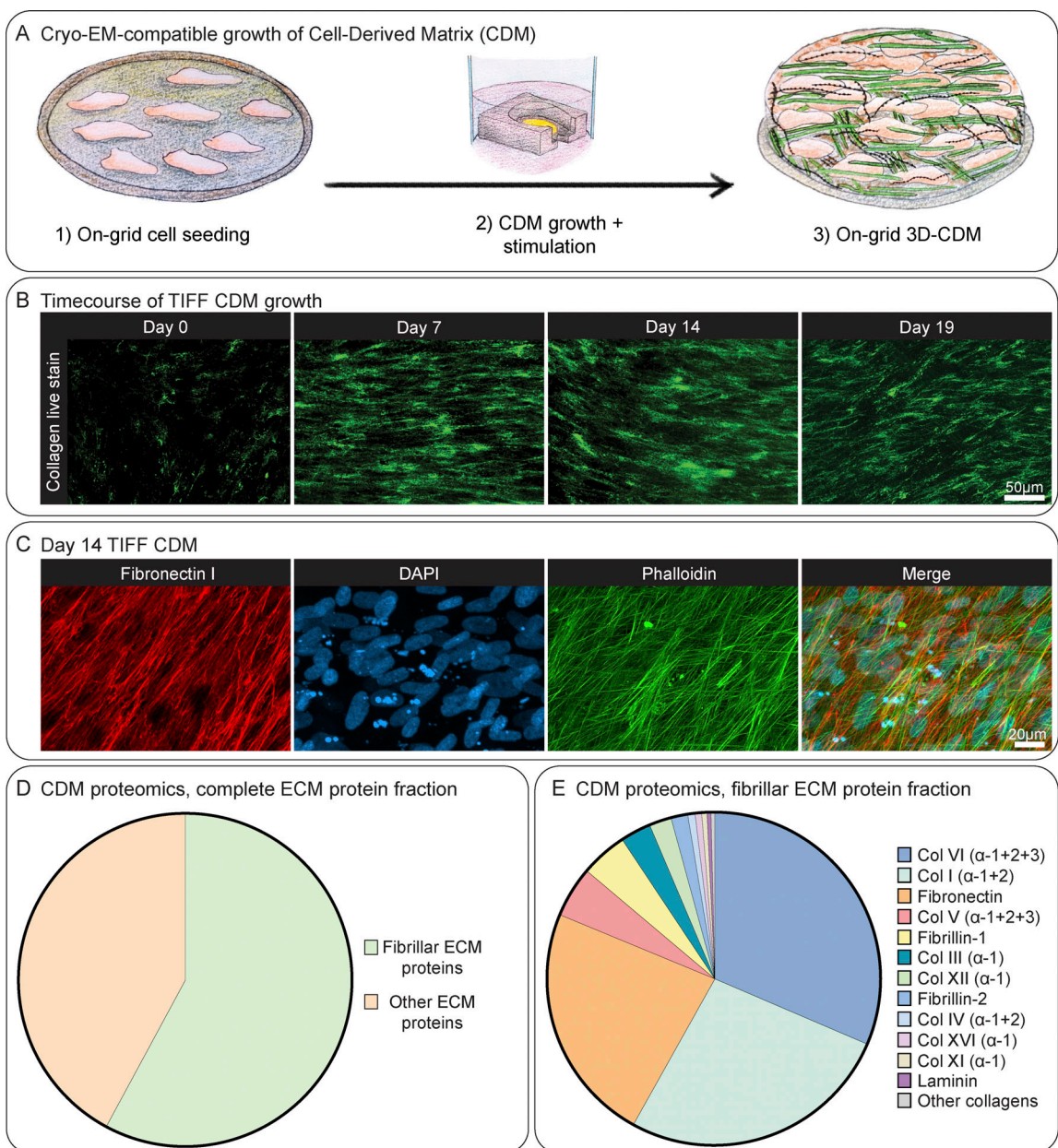

Figure 1. **On-grid CDM generation and characterization. (A)** Schematic depiction of CDM growth on EM grids (see Materials and methods for details). **(B)** Time-course of on-grid collagen fiber growth over 19 days. CDMs were live-stained with the collagen-binding protein CNA35-EGFP and imaged by confocal microscopy. Shown are maximum intensity Z-projections of exemplary CDM areas on different representative EM grids on Day 0, Day 7, Day 14, and Day 19. **(C)** Day 14 TIFF CDMs, fixed, and stained with an anti-FN I antibody, and DAPI and phalloidin to visualize the nucleus and actin cytoskeleton, respectively. An exemplary region acquired by confocal imaging of a CDM is shown as the maximum intensity Z-projection of each staining as well as a merge of all three stainings. **(D and E)** Semiquantitative comparison between different ECM components in Day 14 TIFF CDMs as determined by mass spectrometry. **(D)** A pie-chart comparison of the relative amounts of fibrillar ECM proteins versus other ECM proteins in CDMs. **(E)** A pie-chart comparison of the relative abundance of the different fibrillar ECM proteins found in our CDMs. Scale bar dimensions are shown in the figure.

grid integrity and to define regions of interest (ROI) (Fig. S2 B). We then performed lift-out cryo-FIB milling to obtain CDM-containing lamellae that are thin enough to be subjected to cryo-TEM (Fig. S2 C). Initial vitrification trials without the use of a cryoprotectant resulted not only in lamellae showing high-contrast features in cryo-TEM but also incomplete vitrification (Fig. S3 A). We therefore tested in total 12 different cryoprotectant buffer compositions for overall vitrification, as well as the

additional background they introduced (Table S3). Different cryoprotectant conditions commonly used in other experimental settings (Dahl and Staehelin, 1989; McDonald et al., 2007, 2010; Kaech and Ziegler, 2014; Bharat et al., 2018; Tsang et al., 2018; Borges-Merjane et al., 2020) resulted in insufficient vitrification in all tries (Fig. S3 B). Others resulted in such a high background that cellular and ECM features could not be properly discerned (Fig. S3 C). A degassed cryoprotectant solution containing 10%

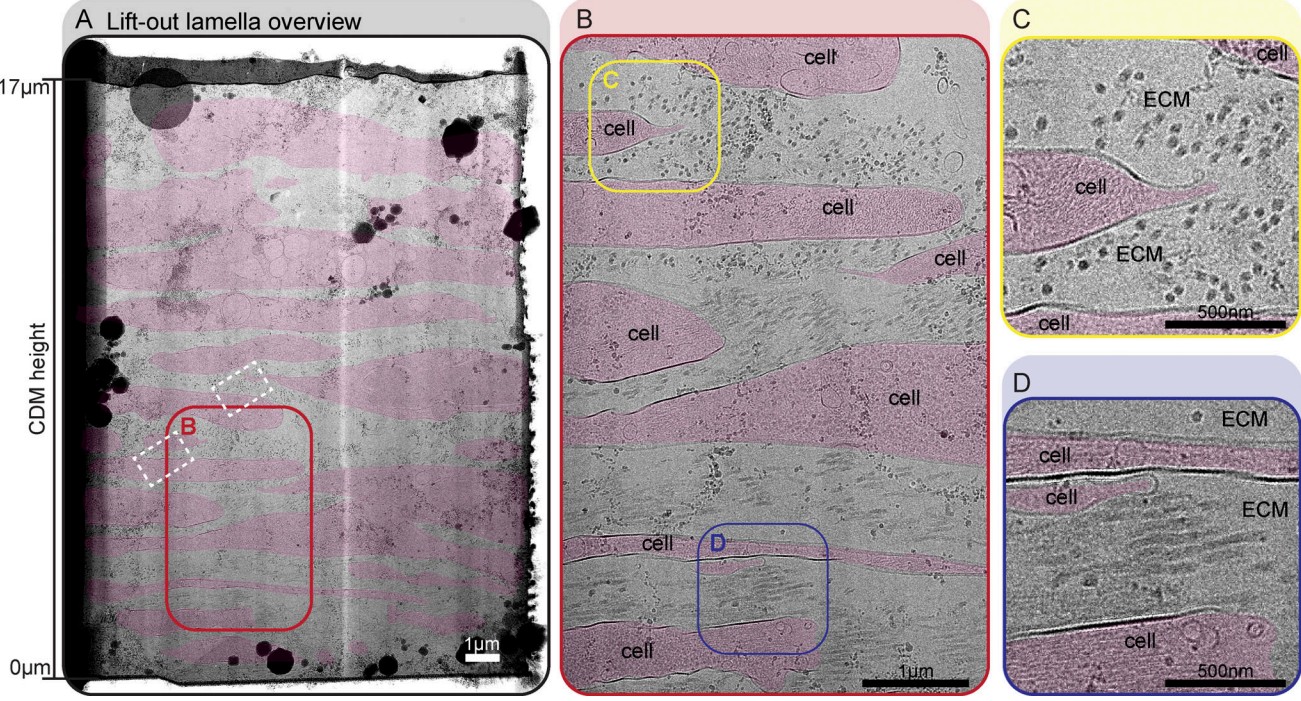

Figure 2. **A fully vitrified cryo-lift-out lamella reveals high-resolution ECM structures. (A–D)** Fully vitrified cryo-lift-out lamella from a TIFF CDM grown for 16 days shown at different magnifications (cryoprotectant: 10% dextran in degassed 0.1 M PB). Cell areas are annotated with transparent red color. **(A)** Complete overview of the cryo-lift-out lamella. The lamella covers roughly 17 µm of CDM depth, ranging from proximal to the EM grid substrate (z = 0 µm) to close to the CDM surface (z = 17 µm). White dashed rectangles denote the areas of acquisition for the tomograms shown in Fig. 3. **(B)** Zoom-in of the lamella as annotated with a red rectangle in A. Two ROIs, highlighted by colored rectangles are shown at higher magnification in C and D. **(C)** Zoom-in into the CDM where ECM fibers are running perpendicular to the lamella, resulting in a cross-section view of fibers. **(D)** Zoom-in into the CDM where ECM fibers are orientated parallel to the lamella, resulting in a side-view of fibers. Scale bar dimensions are annotated in the figure.

(wt/vol) high molecular weight Dextran (40 kD) in 0.1 M phosphate buffer (PB) showed the highest success and resulted in complete vitrification in several samples with low additional background (Fig. 2 and Fig. S4). High MW dextran is a nonpenetrating polymer reported to have little osmotic effect on tissues (Dahl and Staehelin, 1989; Al-Amoudi et al., 2004), resulting in its routine use to facilitate vitrification of biological samples by HPF (Dahl and Staehelin, 1989; Sader et al., 2009; Bharat et al., 2018; Mesman, 2013; Zhang et al., 2021). Hence, we proceeded with this cryoprotectant buffer for our further experiments.

### Architecture of natively preserved CDMs

Cryo-TEM of our fully vitrified lift-out lamellae allowed us to visualize cell–ECM assemblies in a vertical cross-section view spanning over almost the entire depth of the CDM (Fig. 2 and Fig. S4). The CDM was composed of several cell layers, between which extracellular space was filled with ECM components (Fig. 2 B), in line with our observations made by array tomography. Most prominently, we could observe thick ECM filaments running in an orthogonal or parallel direction with respect to the lamella plane, again depending on their position with respect to CDM height. Empty areas in extracellular space devoid of any structures were also observed, and a decrease in ECM fiber density from bottom toward the top of the CDM could be seen in a majority of lamellae.

### Molecular view of natively preserved CDM

We acquired cryo-electron tomograms (n = 43, average thickness = ~194 nm, SD = ± 31.5 nm) from our lift-out lamellae, providing a high-resolution 3D view of the molecular components and the connections between cells and assembled ECM. Our tomograms revealed numerous cellular organelles or membranous compartments, besides hitherto undescribed ECM structures (Fig. 3, Fig. S5, and Videos 1 and 2).

Within cells, we observed cytoskeletal filaments, identified via their characteristic diameters and appearances as intermediate filaments, actin filaments, and microtubules (Fig. 3, A and B; and Fig. 4). Some microtubules showed globular intraluminal microtubule-associated proteins, while other microtubules contained continuous filamentous densities, resembling structures of luminal actin recently observed in HAP60 and *Drosophila* S2 cells (Ventura Santos et al., 2023; Paul et al., 2020). Both ECM fibers, as well as cytoskeletal filaments, displayed a parallel orientation with respect to the long axis of the cell.

Other cellular features included endocytic sites with proteins assembling at the vesicular neck, and fully formed clathrin-coated vesicles, as judged by their dimensions and the clear presence of a protein coat (Fig. S6, A–C). We also observed structures resembling endoplasmic reticulum (ER) compartments (Fig. S6 D) with areas of high local membrane curvature and other membrane-enclosed spherical entities.

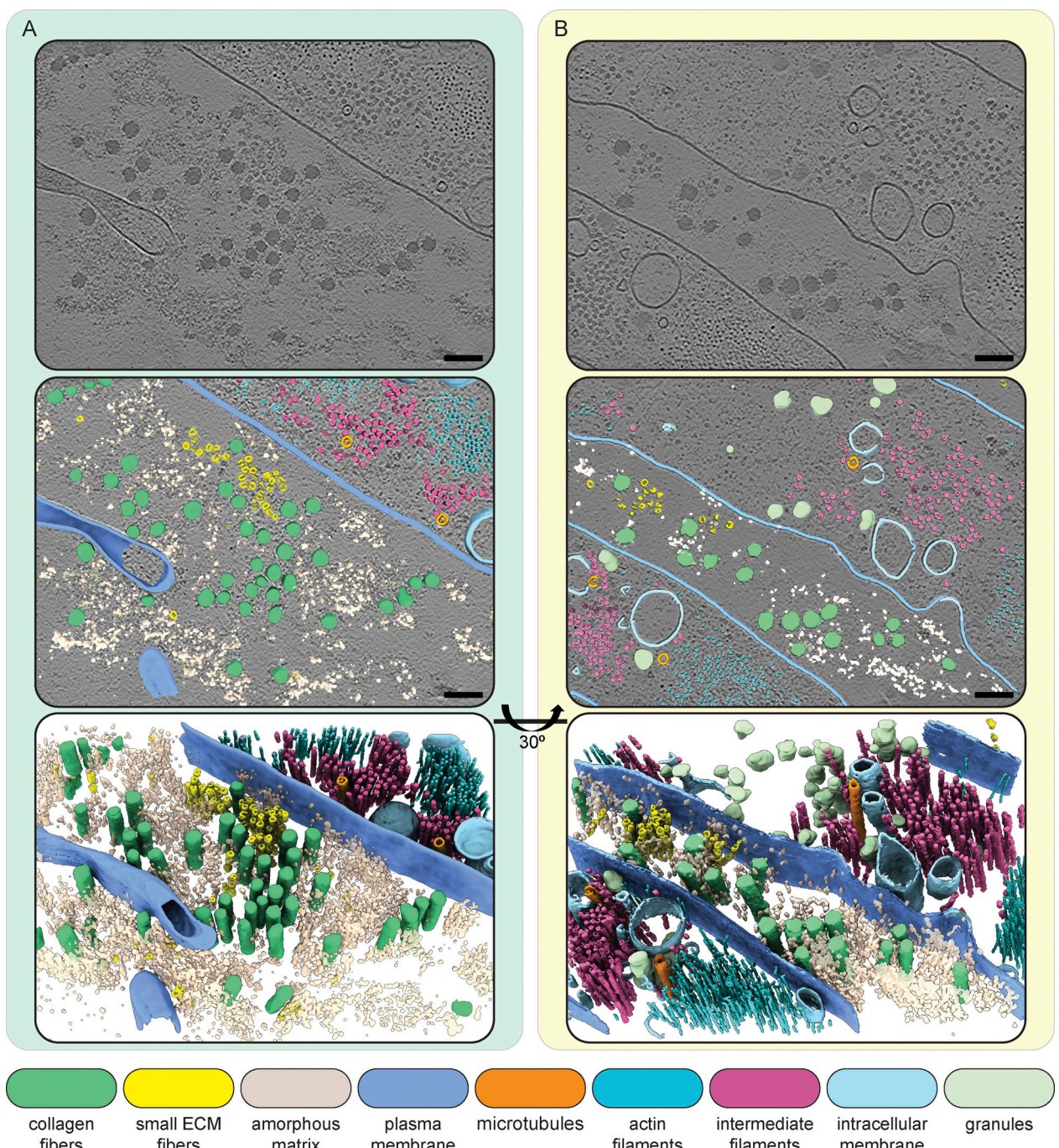

Figure 3. **Molecular landscape of CDMs. (A and B)** Segmentations of two exemplary IsoNet-processed tomograms acquired on the cryo-lift-out lamella shown in Fig. 2. The top panels show a single central slice (1.71 nm thickness) of each tomogram. Cell and ECM fibers are aligned perpendicular to the lamella, resulting in a cross-section view of intra- and extracellular filaments. Middle panels show segmentations of tomograms overlaid over the tomogram slice. The bottom panels show an oblique view of just the segmentation volume. Scale bars indicate 100 nm. The color scheme for the different segmented cellular and ECM components is described in the figure.

**Distinct structural entities within the ECM**

The ECM could be visually classified into four distinct structural entities (Fig. 4). The most abundant and prominent ECM structures were large filaments with a diameter between 25 and 60 nm (Figs. 3 and 4, Fig. S5, and Videos 1 and 2), which we identified as collagen fibers, based on the banding pattern with a periodic D-spacing of ∼67 nm (Fig. S7, A and B). Based on their abundance in our cryo-ET data and proteomics data (Table S2), we assume these to be Col-I fibers. Our resolution does not reveal the triple helix arrangement of collagen but shows them as highly electron dense fibers, tightly embedded in other ECM components.

The second entity consisted of smaller, less regular filament assemblies distributed within the ECM (Figs. 3 and 4). These

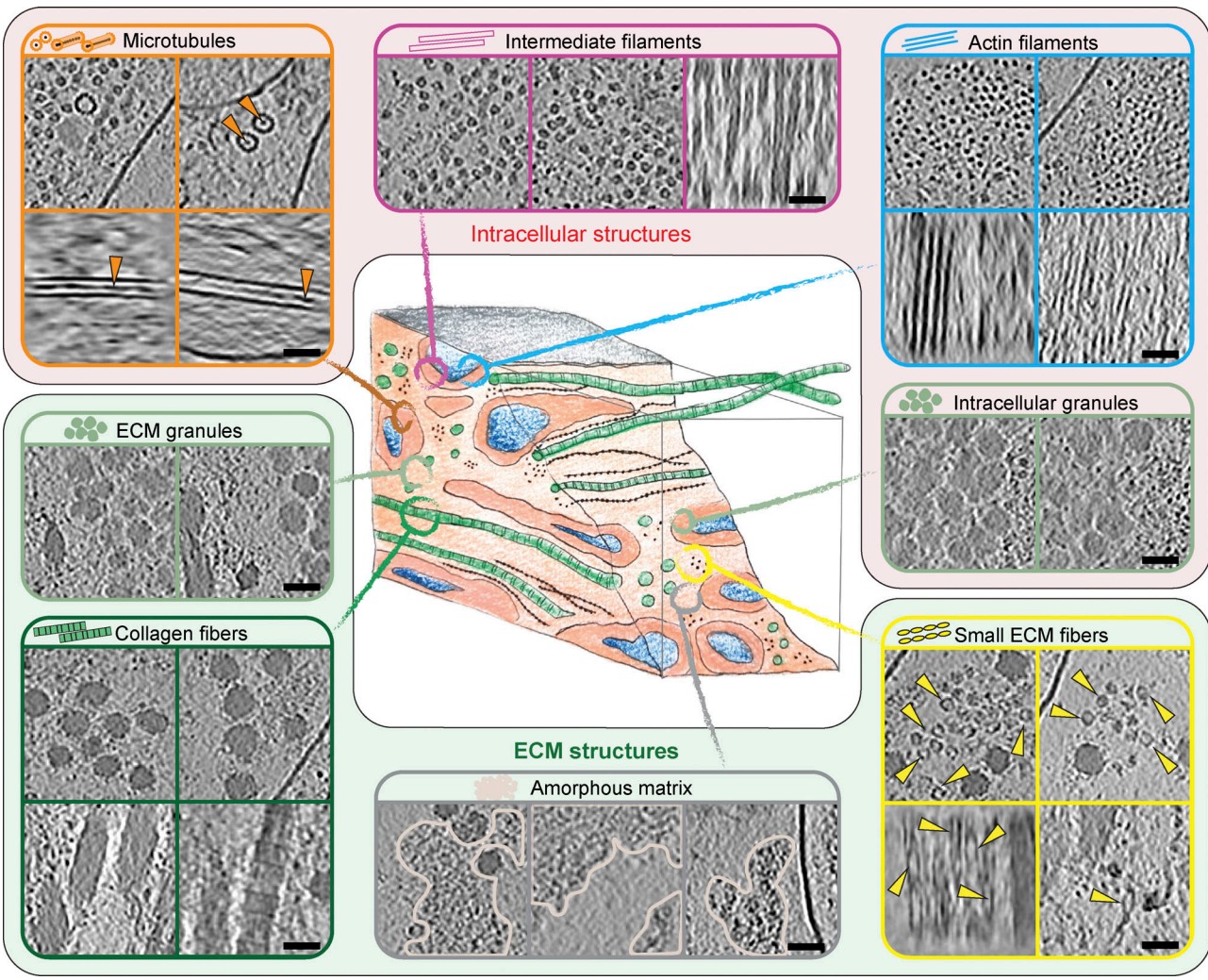

Figure 4. **Gallery of extra- and intracellular structures in CDMs.** Details of extra- and intracellular structures found in CDM tomograms. On the top, intracellular features are shown and the bottom area displays ECM features. Each image is derived from an IsoNet-processed tomogram from a single z-slice (1.71 nm thickness). The scale bars indicate 50 nm.

filaments had a diameter of ~15.7 nm at their widest position ($n$ = 110; SD = ± 1.3 nm) and a bead-like appearance with a node-to-node distance of ~60.3 nm ($n$ = 54; SD = ± 7.8 nm) (Fig. S7, C and D). The variation in both diameter and repeat pattern along the filament z-axis suggests extensibility and assembly variability, matching studies on the properties of both FN and fibrillin (Glab and Wess, 2008; Klotzsch et al., 2009; Sherratt et al., 2001; Dzamba and Peters, 1991). To our knowledge, no similar ECM filament assembly has been yet visualized.

The third structural entity was an amorphous matrix that occupied large areas between filaments (Figs. 3 and 4, and Fig. S8). PGs were described to form a hydrogel-like ground substance (Frantz et al., 2010). In all cases, the amorphous matrix was observed only in close proximity to ECM fibers, often sharply delineating the area filled with ECM components to seemingly empty extracellular space. Based on these observations, it is tempting to speculate that the components in these entities might coassemble. In line with this, it has been previously reported that coordinated presence and interactions

between FP and PGs are required for proper ECM function since disruption or loss of PGs such as decorin leads to abnormal collagen assembly (Danielson et al., 1997; Corsi et al., 2002; Sottile and Hocking, 2002; Chen et al., 2020; Kadler et al., 2008; McDonald et al., 1982; Dzamba et al., 1993; Wenstrup et al., 2004; Saunders and Schwarzbauer, 2019).

The last distinct component of the ECM we found were amorphous granules which appeared in clusters or sometimes also as isolated objects (Fig. S9 A and Video 3). Granules of identical morphology were also often found within cells (Fig. 3 B; and Fig. S9, B and C). Here, they again were either isolated or forming areas largely excluding other cellular components. In a few instances, the release of granules into the extracellular space could also be observed (Fig. S9 D), leading us to speculate that these granules could represent ECM assembly intermediates. Accordingly, Col-VI has been described to first form large intracellular aggregates before secretion and formation of beaded filaments (Cescon et al., 2015). However, the bead-like filaments we observed in our tomograms had dimensions inconsistent

with previous Col-VI reports (Table S1; and Fig. S7, C and D). Instead, fibrillin microfibrils in the resting state with a reported ~56 nm periodicity and ~10–20 nm diameter (Table S1) were closest to our measurements of the beaded filaments of ~60 nm periodicity and ~15 nm diameter (Fig. S7, C and D). Certain ECM filaments, such as Fibrillin microfibrils as well as FN fibers have different extensibility, providing adaptability of tissue. This can result in different measured periodicities depending on the tensile state, providing a potential explanation for the discrepancy between measurements among previous studies and also our data. Still, further experiments are pending for the direct identification of the protein(s) constituting the beaded fibers in our data, and their identity remains unclear.

### The absence of other filament-like structures in CDMs
Most strikingly, our data does not contain any other clearly discernible ECM fiber types, as one would expect based on our proteomics data or previous reports of the filamentous assemblies present within the ECM, such as FN fibers or Col-VI (see also Table S1 for published reports on filament spacing and periodicity) (Baldock et al., 2003; Lansky et al., 2019; Sherratt et al., 2001; Früh et al., 2015). However, within the amorphous matrix, we regularly noticed features seemingly following a linear trajectory, but otherwise not standing out from the disordered matrix environment (Videos 4 and 5). This might suggest that certain ECM fiber assemblies might not take on a highly regular shape and morphology but rather retain a structurally non-distinct shape that blends into the amorphous matrix, potentially due to decoration with PGs or other components. While this conclusion appears inconsistent with previous reports on the structure of Col-VI fibers (Cescon et al., 2015), their apparent absence in our tomograms represents an interesting conundrum, deserving additional studies.

### Assigning identities to the unknown
We acknowledge that our analysis of the ECM is impeded by the major limitation and at the same time potential of in situ cryo-ET, which reveals all molecular components without discrimination. Unambiguous molecular assignments are only possible for structures that are already known (partially defeating the purpose of truly exploratory structural biology), when performing subtomogram averaging to determine protein identity from structure (Schur, 2019) or when using immunogold labeling strategies. ECM filaments represent a challenging target for structure determination approaches due to their variability. Novel image processing tools based on neural networks (for example, Rice et al., 2023) or functional studies using genetic knockouts will be required to extend and annotate the gallery of ECM structures.

### Conclusions
Here, we present to our knowledge the first dedicated cryo-ET study of natively preserved ECM. Our workflow using CDMs allows visualizing hitherto undescribed structures and sets the stage for follow-up studies. This includes the structural and functional characterization of single components and their interactions in natively preserved ECM. Specifically, assuming that changes in matrisome composition of ECMs of different origins are reflected on a structural level, cryo-ET of CDMs allows studying how ECM-specific topologies define tissue homeostasis and underlying cellular behavior. Hence, comparative analysis of CDMs derived from cell types of different origins offer an appealing avenue to obtain a clearer depiction of the role of individual ECM components in defining specialized tissue-specific matrices. Specifically, this could be achieved via an integrative approach combining molecular imaging via cryo-ET and proteomics analysis followed by functionally characterizing the role of specific matrix components using genetic knockout approaches. Together with pharmacological treatment to target specific ECM components, this offers the possibility to manipulate ECM production.

A combination of the present workflow with recently introduced methodologies such as the "Waffle method" (Kelley et al., 2022), the "Serial lift-out" technique (Schiøtz et al., 2023), and montage cryo-ET (Peck et al., 2022; Yang et al., 2023; Chua et al., 2024) could substantially improve the throughput. In turn, with increased dataset sizes, this could potentially grant quantitative insights into CDMs and their components.

## Materials and methods
### Cell culture, CDM growth, and CDM decellularization
Wild type *Homo sapiens* telomerase immortalized foreskin fibroblasts (TIFF, obtained from the lab of Michael Sixt, ISTA) were cultured in Dulbecco's modified Eagle's medium (DMEM GlutaMAX, #31966047; Thermo Fisher Scientific), supplemented with 20% (vol/vol) fetal bovine serum (#10270106; Thermo Fisher Scientific), 1% (vol/vol) penicillin–streptomycin (#15070063; Thermo Fisher Scientific), and 2% (vol/vol) 1 M HEPES (#15630080; Thermo Fisher Scientific).

Cells were incubated at 37°C and 5% $CO_2$ in a cell culture incubator. Phosphate-buffered saline (PBS) used for all sterile cell culture work was purchased from Thermo Fisher Scientific (#20012019).

For CDM growth, the protocol from Kaukonen et al. (2017) was adapted to use EM grids as substrate as follows: 150 or 200 mesh gold holey carbon grids (R 2/2, #N1-C16nAu15-01 and #N1-C16nAu20-01, respectively) were purchased from Quantifoil Micro Tools and glow-discharged for 2 min in an ELMO glow discharge unit (Agar Scientific) prior to cell seeding.

Specimens were handled in a laminar flow hood from here on. EM grids were transferred to the lid of a sterile Falcon cell culture dish (#353004; Corning) with Parafilm stretched over its inside. All treatments described here were performed in this cell culture dish until otherwise stated. Grids were washed 1× with PBS and then coated with 20 µl of 50 µg/ml FN (#11051407001; Sigma-Aldrich) in PBS for 1 h at RT. Subsequently, grids were washed twice with PBS and the FN was cross-linked for 30 min at RT with 20 µl of 1% (vol/vol) glutaraldehyde (#E16220; Science Services) diluted in PBS. After three PBS washes, any remaining glutaraldehyde was quenched with 20 µl of 1 M glycine (#0079.2; Lactan) in PBS for 20 min at RT. Grids were washed once in PBS and twice in cell culture medium and then incubated for at least 15 min with cell culture medium prior to cell seeding.

TIFF cells were seeded onto grids in the cell culture dish in a drop of 20 μl with a density of $3.5 \times 10^5$ cells/ml, resulting in ~7,000 cells per grid. The seeded grids were incubated for 1–2 h in the cell culture incubator. During this incubation, 3D-printed cubic grid holders (Fäßler et al., 2020) (see below) were washed once with PBS and twice with cell culture medium and then incubated in cell culture medium for at least 1 h in a 24-well plate (#92424; TPP). EM grids were then transferred into the grid holders and left there to incubate throughout CDM growth. Once the cells had grown into a confluent cell monolayer, typically within 2–3 days of seeding, the medium was exchanged every other day with a new cell culture medium supplemented with 50 μg/ml ascorbic acid (#A92902; Sigma-Aldrich) and 10 mM HEPES.

Throughout all cell culture experiments, Dumont tweezers, medical grade, style 5 and style 7 were used.

## 3D printing of grid holders
Grid holders are described in detail in Fäßler et al. (2020). Square base grid holders were generated using either a Green-TEC Pro filament (3D Jake) or PETG (Filament PM) with a printing resolution of 0.2 mm layer height for the first layer and 0.15 mm for all additional layers. All printing was done using a 0.4 mm nozzle. Stringing was removed from the grid holders after printing and all holders showing printing errors were discarded. Green-TEC Pro grid holders were washed once with perform classic alcohol EP for 30 min and twice with distilled $H_2O$ and then subsequently autoclaved prior to every use. PETG grid holders were treated identical, but were sterilized using 20 min of UV irradiation instead of autoclaving. Grid holders were reused up to 15 times and stored under sterile conditions until use.

## CNA35-EGFP for CDM live-staining
The bacterial expression vector for pET28a-EGFP-CNA35 was a gift from Maarten Merkx (plasmid # 61603; http://n2t.net/addgene:61603; RRID:Addgene_61603; Addgene) (Aper et al., 2014). BL21 *Escherichia coli* cells were used for expression and induced with 0.1 mM IPTG (#R0393; Thermo Fisher Scientific) at an OD600 of 0.6. The construct was expressed at 37°C for 4 h under constant agitation and cells were then pelleted by centrifugation with 6,000 *g* for 15 min at RT. The cell pellet was resuspended in a freshly prepared buffer containing 20 mM Tris (#9090.3; Lactan), 500 mM NaCl (#P029.2; Lactan), 5% (vol/vol) glycerol (#G5516-500ML; Sigma-Aldrich), 2 μM $ZnCl_2$ (#3533.1; Carl Roth), 1 mM PMSF (#P7626-1G; Sigma-Aldrich), and 1 mM TCEP (#HN95.2; Lactan), pH 8.0. The resuspended cell pellets were snap-frozen in $LN_2$ and stored at –80°C until purification.

Cell lysis was achieved through three cycles of freeze/thaw for 20 min at –80°C and 42°C, respectively. Cell debris was removed by centrifugation at 50,000 *g* for 1 h at 4°C. Subsequently, 10% PEI (#24966-100; Polysciences) was added to a final concentration of 0.3% to precipitate nucleic acid. The sample was centrifuged at 4°C for 10 min at 17,000 *g* and the supernatant was taken over to a new tube. Additionally, ammonium sulfate (#1012115000; Millipore Sigma) was added to a final concentration of 40% to precipitate proteins while stirring

overnight at 4°C. Precipitated proteins were pelleted by centrifugation at 6,000 *g* for 10 min at 4°C and then dissolved in 20 mM Tris, 500 mM NaCl, 2 mM TCEP, and 20 mM imidazole (#56750; Sigma-Aldrich), pH 8.0, while stirring at 4°C for 30 min.

CNA35-EGFP was purified from this solution through the application to a nickel sepharose column, HisTrap FF 1 ml (#17531901; Cytiva). After application, the column was washed with washing buffer (20 mM Tris, 500 mM NaCl, 2 mM TCEP, and 20 mM imidazol, pH 8.0) and then elution buffer (20 mM Tris, 500 mM NaCl, 2 mM TCEP, and 250 mM imidazole, pH 8.0). Fractions containing protein were pooled and dialyzed at 4°C overnight against dialysis buffer (20 mM Tris, 500 mM NaCl, and 0.5 mM TCEP, pH 8.0). Aliquots of the purified protein were flash-frozen in $LN_2$ and stored at –80°C until use.

CNA-EGFP was diluted to a final concentration of 1 μM in cell culture medium for all live staining of CDMs. For staining, CDMs were washed once with the staining solution and then incubated with staining solution for 1–2 h in the cell culture incubator. After staining, CDMs were washed three times with cell culture medium, kept in the cell culture incubator, and used for imaging and/or HPF within the next 1–3 h.

## Antibodies and stainings
For fixing and permeabilization of CDMs, we followed the protocol described in Franco-Barraza et al. (2016).

Specimens were recovered from the grid holders and placed on Parafilm stretched over the inside of a cell culture dish lid, after which they were incubated at RT for 5 min in fixing and permeabilization solution (4% PFA [#E15710; Science Services], 4% sucrose [wt/vol, #84100-1KG; Sigma-Aldrich], and 0.5% Triton-X100 [#T8787; Sigma-Aldrich]). Samples were then washed once with fixing solution (4% PFA and 4% sucrose [wt/vol] in PBS) followed by a 20-min incubation in fixing solution at RT. After washing three times gently with PBS, specimens were incubated in blocking solution (3% BSA [#10735078001; Sigma-Aldrich] in PBS) for 1 h at RT. Subsequently, the blocking solution was removed and the sample was incubated in an anti-FN I antibody from rabbit (#F3648; Sigma-Aldrich), diluted 1:500 in blocking solution, overnight at 4°C in a wet chamber.

The next morning, specimens were washed three times with PBS before incubation in a solution of anti-Rabbit-IgG-ATTO 594 (#77671-1ML-F; Merck) as secondary antibody, phalloidin-ATTO 488 (#AD488-81; ATTO-TEC), and 4′,6-Diamidino-2-phenyl-indole dihydrochloride (#32670-5MG-F, DAPI; Sigma-Aldrich), all diluted 1:500 in a blocking solution. Samples were incubated for 1–2 h at RT in a wet chamber in the dark. After this incubation, samples were washed three times with PBS and stored at 4°C in a wet chamber in the dark until imaging.

## Light microscopy
Specimens stained with EGFP-CNA35 were placed on Parafilm in a drop of cell culture medium and quickly assessed at RT for sample integrity and quality prior to vitrification. Fixed specimens stained with antibodies and other dyes were kept on Parafilm in a drop of PBS. For their assessment, we used a Zeiss Axio Imager.Z2 Upright LSM800 microscope equipped with a

Plan-Apochromat 20×/NA 1.0 W DIC water-dipping (WD = 1.8 mm) objective and GaAsP PMT photomultiplier tube detectors. Z-stacks with 1 μm steps over the whole height of the specimen, from the EM substrate to the top of the CDM, were acquired at RT using the ZEN 2.6 software. Typically, at least three positions per specimen were acquired. Live samples were kept at 37°C for the duration of the imaging process and returned to a cell culture incubator after a maximum of 30 min. Alternatively, a Zeiss Axioscope with a W N-Achroplan 20×/0.5 water-dipping (WD = 2.6 mm) objective was used at RT for quick visual assessments of CDM quality prior to HPF.

A maximum intensity Z-projection was applied to Z-stacks using Fiji (Schindelin et al., 2012). To improve the visibility of these images, contrast and brightness were adjusted as necessary.

During this initial assessment, any specimens showing distortions of ECM fibers, damage to the CDM, or bending of the EM grids were removed from the sample pool.

## Mass spectrometry
### Sample processing
TIFF cells were seeded on 10-cm diameter cell culture dishes (#83.3902; Sarstedt) and treated as described above for CDM growth. Cell culture dishes were not coated with FN prior to cell seeding to prevent the introduction of a bias to the mass spectrometry analysis. CDMs were grown for 14 days with ascorbic acid treatment every other day.

CDMs were then decellularized following the protocol published by Kaukonen et al. (2017). Briefly, CDMs were washed with extraction buffer consisting of 0.5% Triton-X100 and 0.56–0.6% $Na_4OH$ (#221228-1L-A; Sigma-Aldrich) in PBS that had been prewarmed to 37°C. The washes were repeated until the fibroblasts were extracted, as observed by phase contrast microscopy on a standard cell culture stereo microscope. Typically, this process takes up to 5 min and three to four washes of TIFF CDMs. After cells were extracted, CDMs were washed with PBS and treated for 1 h at 37°C with a buffer of 50 μg/ml DNaseI (#11284932001; Roche), 5 mM $MgCl_2$, and 1 mM $CaCl_2$ diluted in PBS. Specimens were then washed three times with PBS and immediately fixed with 4% PFA in PBS. After decellularization, CDMs were scraped off the cell culture dish surface with a cell scraper (#83.1830; Sarstedt) and the matrix was transferred into 1.5-ml centrifugation tubes.

Adapting the protocol published in Lansky et al. (2019), CDMs were centrifuged (13,000 $g$, 5 min), supernatants were removed, and the resulting pellets were resuspended in a buffer consisting of 100 μl 8 M urea (#U1250; Sigma-Aldrich), 100 mM TEAB (triethylammonium bicarbonate; #T7408-100ML; Sigma-Aldrich), and 25 mM TCEP (tris(2-carboxyethyl)phosphine hydrochloride, #77720; Thermo Fisher Scientific). Samples were then sonicated (Bioruptor plus, Diagenode, 10 × 30 s/30 s ON/OFF cycles) and heated up to 37°C for 2 h while shaking (800 rpm, Thermomixer F1.5, Eppendorf). Following this, specimens were alkylated by the addition of 100 μl 50 mM iodoacetamide (#A39271; Thermo Fisher Scientific) and incubated for 30 min in the dark while shaking (800 rpm). Then, 200 μl 100 mM TEAB was added to the specimens and they were digested with 10 μl PNGase (#A39245; Gibco) at 50°C for 2 h.

The sample was diluted by the addition of 390 μl 100 mM TEAB, then supplemented with 8 μl Trypsin/LysC (1 μg/μl; #V5072; Promega) and digested overnight at 37°C. Following this, 4 μl Trypsin/LysC (1 μg/μl) was added to the sample and it was incubated for a further 2 h, then 90 μl 10% TFA (trifluoroacetic acid; #10723857; Thermo Fisher Scientific) was added for acidification. A tC18 SepPak plate (#1860002318; Waters) was used for clean-up according to the manufacturer's protocol.

### LC-MS/MS analysis
The sample was dried, redissolved in 0.1% TFA, and analyzed by LC-MS/MS on an Ultimate high-performance liquid chromatography (nano HPLC, Thermo Fisher Scientific) coupled to a Q-Exactive HF (Thermo Fisher Scientific). An Acclaim PepMap C18 trap-column (5 μm particle size, 0.3 mm ID × 5 mm length; #160454; Thermo Fisher Scientific) was used to concentrate the sample, which was then bound to a 200 cm C18 μPAC column (micro-Pillar Array Column; #5525031518210B; PharmaFluidics) and finally eluted with a constant flow of 600 nl/min over the following gradient: solvent A, 0.1% formic acid (#160454; Thermo Fisher Scientific) in water; solvent B, 80% acetonitrile (#10001334; Thermo Fisher Scientific) and 0.08% formic acid in water. Step 1: 5 min of 2% solvent B. Step 2: 160 min of 31% solvent B. Step 3: 185 min 44% solvent B. Mass spectra were acquired in positive mode with a data-dependent acquisition method: full width at half maximum (FWHM) 120 s, mass spec scan acquired without fragmentation parameters (MS1): profile, 1 microscan, 120,000 resolution, automatic gain control (AGC) target 3e6, 50 ms maximum IT, 380–1,500 m/z; up to 20 MS2s per cycle. Mass spec scan acquired after one round of fragmentation (MS2) parameters: Centroid mode, 1 microscan, 15,000 resolution, AGC target 1e5, 20 ms maximum IT, 1.4 m/z isolation window (no offset), 380–1,500 m/z, NCE 28, excluding charges 1+, 8+ and higher or unassigned, and 60 s dynamic exclusion.

### Data analysis
Raw files were searched in MaxQuant 1.6.17.0 against a *H. sapiens* reference proteome downloaded from UniProtKB. Fixed cysteine modification was set to carbamidomethyl. Variable modifications were oxidation (M), acetyl (Protein N-term), deamidation (NQ), Gln->pyro-Glu, Phospho (STY), and hydroxyproline. Match between runs, dependent peptides, and second peptides were active. All false discovery rates (FDRs) were set to 1%. MaxQuant results were further processed in R using in-house scripts, which, starting from MaxQuant's evidence.TXT (PSM) table, perform parsimonious protein groups inference and generate an Excel-formatted protein groups table. GO annotations were downloaded from UniProtKB. ECM proteins were defined as proteins annotated with the GO term "Extracellular Matrix" (GO:0031012) or any of its descendants.

ECM proteins were then sorted according to the normalized $\log_{10}$ of the estimated protein group expression value and listed in Table S2.

### Array tomography
Sample preparation was done according to the OTO fixation protocol described in Deerinck et al. (2010) to enhance the contrast of the sample for SEM.

TIFF CDMs were grown on glass coverslips (#CB00150RA-120MNZ0; Epredia) for 14 days with ascorbic acid treatment as described in Cell culture, CDM growth, and CDM decellularization, and then fixed with 2% PFA and 2.5% glutaraldehyde in 0.1 M PB for 1 h at RT. Samples were washed with 0.1 M PB, and contrast was enhanced using 2% aqueous osmium tetroxide (#E19110; Science Services) and 1.5% potassium ferrocyanide (#P9387-100G; Sigma-Aldrich) in 0.1 M PB for 30 min in the dark. After washing with MilliQ water, the samples were incubated in thiocarbohydrazide (#88535-5G; Sigma-Aldrich) for 20 min at RT and subsequently washed with MilliQ water.

Following this, specimens were placed in 2% aqueous osmium tetroxide for 30 min at RT in the dark and then washed again with MilliQ water. Samples were then incubated overnight in 1% aqueous uranyl acetate (#77870.02; AL-Labortechnik) at 4°C. The following morning, they were again washed with MilliQ water, incubated in Walton's lead aspartate solution prepared from L-aspartic acid (#A8949-25G; Sigma-Aldrich) and lead nitrate (#228621-100G; Sigma-Aldrich) for 30 min at 60°C, and washed again in MilliQ water.

Samples were subjected to a graded series of ethanol of 50%, 70%, 90%, and 2 × 100% (#32221-2.5L; Bartelt) for dehydration and then placed in anhydrous acetone (#CL0001722500; Bartelt). They were then infiltrated in a graded series of hard DurcupanTM ACM resin (Component A: #44611-100ML, B: #44612-100ML, C: #44613-100ML, D: #44614-100ML; Sigma-Aldrich) in acetone and placed in pure Durcupan overnight. The following day, the coverslips were put on an ACLAR foil (#E50425-10; Science Services) and a BEEM capsule (size 00, #E70020-B; Science Services), filled with fresh resin, and placed upside down in the middle of each coverslip. Samples were placed in a 60°C oven for 3 days for polymerization. After this, they were dipped in LN₂ until the coverslip could be carefully removed with a razor blade.

Samples were trimmed with an Ultratrim diamond knife (Diatome) using an Ultramicrotome EM UC7 (Leica Microsystems). A carbon-coated 8 mm wide Kapton tape (RMC Boeckeler) was plasma-treated using an ELMO glow discharge unit, equipped with a homemade reel-to-reel motorized winder, to increase its hydrophilicity. Serial ultrathin sections of 70 nm thickness were cut with a 4 mm Ultra 35 diamond knife (Diatome) and lifted up with the plasma-treated tape using an automated tape-collecting Ultramicrotome ATUMtome (RMC Boeckeler).

The tape used to collect the serial sections was cut into strips and mounted on a 4-inch silicon wafer (University Wafer) with conductive double-sided adhesive carbon tape (#P77819-25; Science Services). The wafer was then coated with a 5 nm carbon layer using an EM ACE 600 (Leica Microsystems) to ensure conductivity. The collected sections were then imaged on a Field Emission-SEM Merlin compact VP (Carl Zeiss) equipped with the Atlas 5 Array Tomography software. The high-resolution serial images for 3D-SEM reconstruction were taken with 10 nm pixel resolution at 5 kV using a backscattered electron detector.

Serial SEM images were downsampled to approximately isotropic resolution (x,y: 80 nm, thickness 70 nm). These images were then aligned using a custom MATLAB script: The optimal affine transformation linking consecutive images was found

employing an evolutionary optimizer based on pairwise SURF features, employing the M-estimator sample consensus algorithm and using mean squares as a quality metric.

The process of pixel classification for cell bodies, nuclei, and ECM was executed using the auto-context workflow in Ilastik (version 1.4.0) (Berg et al., 2019). The classification of filamentous structures was done separately via the pixel classification workflow in Ilastik. The output generated was subsequently imported into Imaris (version 9.3) for visualization and reconstruction of the cell body and nuclear surfaces as shown in Fig. S1 C.

To visualize the alignment of the cell/nucleus major axis with the fiber orientation, the Fiji plugin OrientationJ was used (settings: σ = 16, gradient = Cubic Spline) (Püspöki et al., 2016) as shown in Fig. S1 D.

### High-pressure freezing (HPF)
Carriers were designed to fit the Z-height of on-grid CDMs and custom-produced at the ISTA Miba Machine Shop. Two types of 3 mm diameter carriers were combined for HPF of CDMs: Carriers of type A had a height of 0.5 mm, with a 2 mm diameter recess of a depth of ∼20 μm (±5 μm machining inaccuracy). Carriers of Type B had a height of 0.5 mm without any recess. To ensure proper carrier sandwich height after assembly, every single carrier was measured manually with a digital Vernier caliper for its height, and any carrier with more than ±20 μm derivation in height was removed.

All carriers were cleaned by three rounds of sonication in pure ethanol and then dried at 60°C on a hot plate. Prior to use, carriers were fully coated with 1-hexadecene (#H2131-100ML; Sigma-Aldrich). CDMs were incubated in the used cryoprotectants listed in Table S3 30 min prior to vitrification and kept at 37°C, 5% CO₂ during this incubation time. CDMs were kept at 37°C up until HPF carrier sandwich assembly and then frozen as quickly as possible with a BAL-TEC HPM010. To avoid excess contamination, specimens were stored in cryo-vials in LN₂ directly after HPF and then transferred to a freshly cooled, clean clipping station for disassembly. Recovered specimens were clipped into FIBSEM AutoGrids (Thermo Fisher Scientific) marked as described in Wagner et al. (2020) and stored until further use.

The cryoprotectants listed in Table S3, dextran (MW 40 kD, #31389-100G), sucrose (#84100-1KG), polyvinylpyrrolidone (PVP, #PVP10-100G), and BSA (#10735078001), were purchased from Sigma-Aldrich.

### Cryofluorescence microscopy
All specimens were screened on a Leica EM Cryo CLEM microscope (Leica Microsystems) using the Leica Application Suite 3.7.0. The LasX navigator was used to acquire tile scans of entire FIBSEM Autogrids to facilitate correlation for FIB milling. Z-stacks of regions of interest were acquired to assess the presence of collagen fibers and to select the best positions for ion-beam milling. All specimens showing damage to the CDM or strong distortions of the EM grid were discarded after imaging.

### Cryo-FIB milling
Cryo-lift-out lamellae were generated using a second-generation Aquilos (Aquilos II) instrument (Thermo Fisher Scientific). The

instrument was operated using the xT user interface and the MAPS 3.14 software (TFS). The FIB was operated at 30 kV, and the milling progress was monitored using the SEM beam at 25 pA and 2–5 kV.

Prior to sample loading, a half-moon grid with four finger-like extensions for lift-out attachment (#10GC04; Ted Pella) was clipped into an AutoGrid. For clipping, all finger-like extensions of the half-moon grid were oriented in line with the milling window to allow for low-angle sample thinning.

After loading sample and half-moon grid, overview maps of the high-pressure frozen specimens were acquired and correlated with the images obtained on the Leica EM Cryo CLEM microscope to identify regions of interest. The milling slot on the FIBSEM AutoGrid was used for improving correlation by using its rim, visible in both reflected light microscopy and SEM, as a landmark.

Then specimens were sputter-coated with platinum for 30 s at 30 mA and 10 Pa with the built-in sputter coater, followed by a GIS deposition of 1.5–2 μm metalorganic platinum. Another tile-scan overview image was then taken using the MAPS software.

Lift-out sites were identified by CLEM and set to eucentric position. Steps performed for the lift-out FIB milling will be explained using Fig. S2 C as illustration. Fig. S2 C-(1): Trenches for the lift-out procedure were cut at a stage tilt of 7° and a relative stage rotation of 180° to position the FIB perpendicular to the sample. The trenches in front, behind, and to the side of the lift-out were milled in cross-section (CS) patterns with 3 nA, and their size was adjusted as needed. The front and back of the lift-out were polished smooth by milling with cross-section-cleaning (CSC) patterns fitted to the width of the lift-out with 1 nA. Fig. S2 C-(2): Undercuts were performed at 28° stage tilt, at the default stage rotation, with 1 nA. Rectangle milling patterns were placed below and to either side of the lift-out, leaving it attached to the bulk sample only on a small anchor, marked red. The micromanipulator needle was then attached by redeposition milling, using a CS pattern below and above the needle set to 0.5 μm z-depth, with a Multi-Pass setting of 1 at 0.5 nA. Pattern placement is shown in blue in the figure. After visual confirmation of successful attachment, the remaining anchor to the bulk sample was removed at 0.5 nA with a rectangle pattern placed at the anchor position, as indicated by the red pattern in the figure. Figs. S2 C-(3) and S2 C-(4): The lift-out was then lifted up and out of the bulk sample and transferred to the second shuttle position, which held the half-moon grid. The fingers of the half-moon grids were prepared for attachment by milling the attachment site to be flat using 5 nA at perpendicular ion beam position with CS pattern. Fig. S2 C-(5): The lift-out was attached to a finger by redeposition milling, using four CS patterns at 0.5 nA with a Multi-Pass setting of 1, a z-depth of 3 μm, and a surface of ∼2 × 2 μm. The pattern placement is indicated in blue in the figure. Following visual confirmation of the attachment, the needle was pulled off gently to the side, so it could be directly reused for the next lift-out without any necessary cleaning steps. Fig. S2 C-(6): The portion of the lift-out that was used for the needle-attachment was then removed by placing a rectangle milling pattern and FIB milling with 1 nA, as indicated by a red pattern in the figure. Fig. S2 C-(7) to S2 C-(9): Each lift-out was then milled down with rectangle patterns to ∼200–250

nm thickness in several steps, reducing lamella width and milling current in each step, resulting in a symmetric stair-like anchor. Lamellae were thinned down to 3 μm thickness with 1 nA, then to 1.5 μm thickness with 0.5 nA, and to 900 nm using 0.1 nA. Here, the stage was tilted to ±1° and the lamella overhangs above and below were milled with 0.1 nA to facilitate a parallel shape of the lamella rather than a wedge-like one. The lamellae were then milled down to 500 nm with 50 pA and then again lamella overhangs were removed with ±0.5° tilts. In a final step, each lamella was thinned down to 200 nm with a milling current of 30 pA.

All samples were stored in autogrid boxes in liquid nitrogen until TEM imaging.

### Cryo-TEM and cryo-ET

A TFS Titan Krios G3i operated at 300 kV in nanoProbe energy-filtered transmission electron microscopy (EFTEM) mode equipped with a Gatan K3 BioQuantum direct electron detector with a slit width of 20 eV was used for data acquisition on cryo-lift out lamellae. Zero Loss Peak (ZLP) and energy filter tuning were done using DigitalMicrograph 3.42 (Gatan). Coma-free alignment and astigmatism correction were performed using SerialEM 4.0 beta5 (Mastronarde, 2003). For medium magnification images, a pixel size of 13.74 Å at a nominal magnification of 6,500× was used.

For tilt series acquisition, the camera was operated in counting mode using hardware binning and dose fractionation, with eight frames per tilt. The total dose applied was 180 e/Å², divided into 61 images (for a 2° increment tilt scheme). All tilt series were acquired with a dose-symmetric scheme starting from the lamella pre-tilt angle in a range of –60° and +60° (Hagen et al., 2017) at a defocus of –8 μm. The pixel size was set to 2.137 Å at a nominal magnification of 42,000×. Datasets were acquired using SerialEM (Mastronarde, 2005) and PACE-tomo (Eisenstein et al., 2022, *Preprint*).

Tomograms were reconstructed using weighted back-projection and patch tracking, employing the IMOD software (Kremer et al., 1996) as well as AreTomo (Zheng et al., 2022), with a binning of 8. Bad tilts compromised by movement, beam obstruction, or strong reflections were removed. A SIRT-like filter (equivalent to 15 iterations) was applied during tomogram reconstruction for selected tomograms to visualize the collagen banding pattern more clearly, as shown in Fig. S7 A.

Tomograms were additionally processed with IsoNet (Liu et al., 2022) to increase the signal-to-noise ratio (SNR) and fill in the missing wedge information. For this, raw bin 8 tomograms were deconvoluted with IsoNet and subsequently used to train the neural network for 50 iterations in batches of 10 tomograms. The mask patch size used for training was set to six subtomograms with a cube size of 64 pixels. The same tomograms were then also used for the reconstruction of the missing-wedge information and improvement of the SNR. All data from tomograms shown in this paper originate from these IsoNet-filtered tomograms unless otherwise stated.

### Segmentation

Tomograms were segmented in the Dragonfly software, Version 2020.2 for Linux (Object Research Systems Inc., 2020). For each

tomogram, four slices were selected in the Segmentation Wizard and manually segmented. These segmented slices were then used for the training of a deep learning model utilizing a U-Net architecture, with an input dimension of 2.5D (number of slices = 3), a depth level of 5, and an initial filter count of 128. Following this training, the generated model was used to segment the whole tomogram and corrections were manually applied as necessary. Each class of object was extracted as an ROI from the segmentation and rendered into a 3D contour mesh. Following this, the mesh was smoothened using the Laplacian Smoothing method with two to five iterations as needed, and the smoothened contour mesh was exported as a.stl file for visualization in UCSF ChimeraX (Pettersen et al., 2021).

The volume shown in Fig. 3 A was partially segmented in the Amira-Avizo software, version 2020.2 (Thermo Fisher Scientific): Plasma membranes and vesicles were tracked manually in Amira. All manually segmented structures were exported from Amira as.mrc files and then imported into ChimeraX, smoothened, and visualized together with the files imported from Dragonfly. All objects were rendered for display using ChimeraX.

To display all objects on the same scale in ChimeraX, Dragonfly files were first upscaled to match the tomogram dimensions. Where there were obvious defects from Dragonfly export, model meshes were manually fixed using Blender 3.5.1. All models were loaded into ChimeraX, and the positions of.stl objects were manually positioned relative to the tomogram. Amorphous density in the extracellular matrix was generated by first inverting density for each tomogram and then using the "Segger" tool in Chimera 1.17.1 (Pettersen et al., 2004). Regions of density inside of cells were manually excluded, and the remaining density in the ECM was extracted from the volume for rendering in ChimeraX.

Final figures and movies were generated using ChimeraX. Camera perspective was set to mono, and lighting depthCueStart and depthCueEnd were both set to 1. Lighting and silhouettes were otherwise set to default settings using the "soft" lighting preset. A volume box outline was colored "grey" and displayed for the amorphous matrix only. The amorphous matrix was thresholded manually to yield a result representative of the raw tomogram and Gaussian-filtered to a value of 22 within Chimera 1.17.1. The amorphous matrix was colored #FFDAB9 (peach puff) with transparency set to 40% for images and 0% for movies. Plasma membranes were colored #6495ED (cornflower blue), vesicles were colored #87CEEB (sky blue), collagen was colored #3CB371 (medium sea green), small ECM filaments were colored #FFFF00 (yellow), and microtubules were colored #FF8C00 (dark orange). Actin was colored #00F0F0 (cyan), intermediate filament was colored #DC5A9B (magenta), and granules were colored #8FB38D (light green). Tomogram slices were overlaid with the segmentation using ArtiaX version 0.3 plugin (Ermel et al., 2022). Slice representation was set to "plane" in the volume viewer. Tomograms were manually brightened within ArtiaX for display purposes.

Figures were assembled in Adobe CC Illustrator (2023).

## Online supplemental material

Fig. S1 shows room-temperature SEM array tomography of TIFF CDMs grown for 14 days. Fig. S2 shows HPF and cryo-lamella preparation of CDMs for cryo-ET. Fig. S3 shows examples of incompletely vitrified CDM lift-out lamellae. Fig. S4 shows additional examples of completely vitrified cryo-lift-out lamellae. Fig. S5 shows additional examples of segmented cryo-electron tomograms. Fig. S6 shows membrane compartments—endocytic sites, clathrin-coated vesicles, and ER compartments. Fig. S7 shows details of ECM fibers. Fig. S8 shows amorphous matrix details. Fig. S9 shows extra- and intracellular ECM-associated granules. Video 1 is a video of the tomogram and segmentation shown in Fig. 3 A. Video 2 is a video of the tomogram and segmentation shown in Fig. 3 B. Video 3 is a video of the tomogram and segmentation show in Fig. S9 D. Video 4 is a video showing features following a linear filament-like trajectory. Video 5 is a video showing features following a linear filament-like trajectory. Table S1 lists examples of reported dimensions for different ECM fibers. Table S2 lists predominant ECM proteins in TIFF CDMs identified by mass spectrometry. Table S3 presents a scouting of different cryoprotectants and buffers for their vitrification potential.

## Data availability

Representative tomograms containing the data shown in Fig. 3, and Figs. S5 and S9 have been deposited in the Electron Microscopy Data Bank under accession codes: EMD-18490, EMD-18491, EMD-18492, EMD-18493, and EMD-18494. All tilt series, including frames, raw tilt series, and reconstructed IsoNet-filtered tomograms, were deposited in the Electron Microscopy Public Image Archive database under the accession code EMPIAR-11897.

## Acknowledgments

We thank Armel Nicolas and his team at the ISTA proteomics facility, Alois Schloegl, Stefano Elefante, and colleagues at the ISTA Scientific Computing facility, Tommaso Constanzo and Ludek Lovicar at the Electron Microsocpy Facility (EMF), and Thomas Menner at the Miba Machine shop for their support. We also thank Wanda Kukulski (University of Bern) as well as Darío Porley, Andreas Thader, and other members of the Schur group for helpful discussions. Matt Swulius and Jessica Heebner provided great support in using Dragonfly. We thank Dorotea Fracciolla (Art & Science) for support in figure illustration.

This research was supported by the Scientific Service Units of ISTA through resources provided by Scientific Computing, the Lab Support Facility, and the Electron Microscopy Facility. We acknowledge funding support from the following sources: Austrian Science Fund (FWF) grant P33367 (to F.K.M. Schur), the Federation of European Biochemical Societies (to F.K.M. Schur), Niederösterreich (NÖ) Fonds (to B. Zens), FWF grant E435 (to J.M. Hansen), European Research Council under the European Union's Horizon 2020 research (grant agreement No. 724373) (to M. Sixt), and Jenny and Antti Wihuri Foundation (to J. Alanko). This publication has been made possible in part by CZI grant DAF2021-234754 and grant DOI https://doi.org/10.37921/812628ebpcwg from the Chan Zuckerberg Initiative DAF, an advised fund of Silicon Valley Community Foundation (to F.K.M. Schur).

Author contributions: Supervision and funding acquisition: M. Sixt and F.K.M. Schur; project administration: B. Zens and F.K.M. Schur; conceptualization: B. Zens and F.K.M. Schur; methodology: B. Zens and F.K.M. Schur; investigation: B. Zens, R. Hauschild, J.M. Hansen, J. Datler, J. Alanko, F. Fäßler, V.-V. Hodirnau, and V. Zheden; validation and formal analysis: B. Zens and F.K.M. Schur; visualization: B. Zens, J.M. Hansen, J. Datler, and R. Hauschild; data curation: B. Zens and F.K.M. Schur; writing—original draft: B. Zens and F.K.M. Schur; writing—review and editing: B. Zens, F. Fäßler, J. Datler, V.-V. Hodirnau, R. Hauschild, V. Zheden, J.M. Hansen, J. Alanko, M. Sixt, and F.K.M. Schur.

Disclosures: The authors declare no competing interests exist.

Submitted: 26 September 2023

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

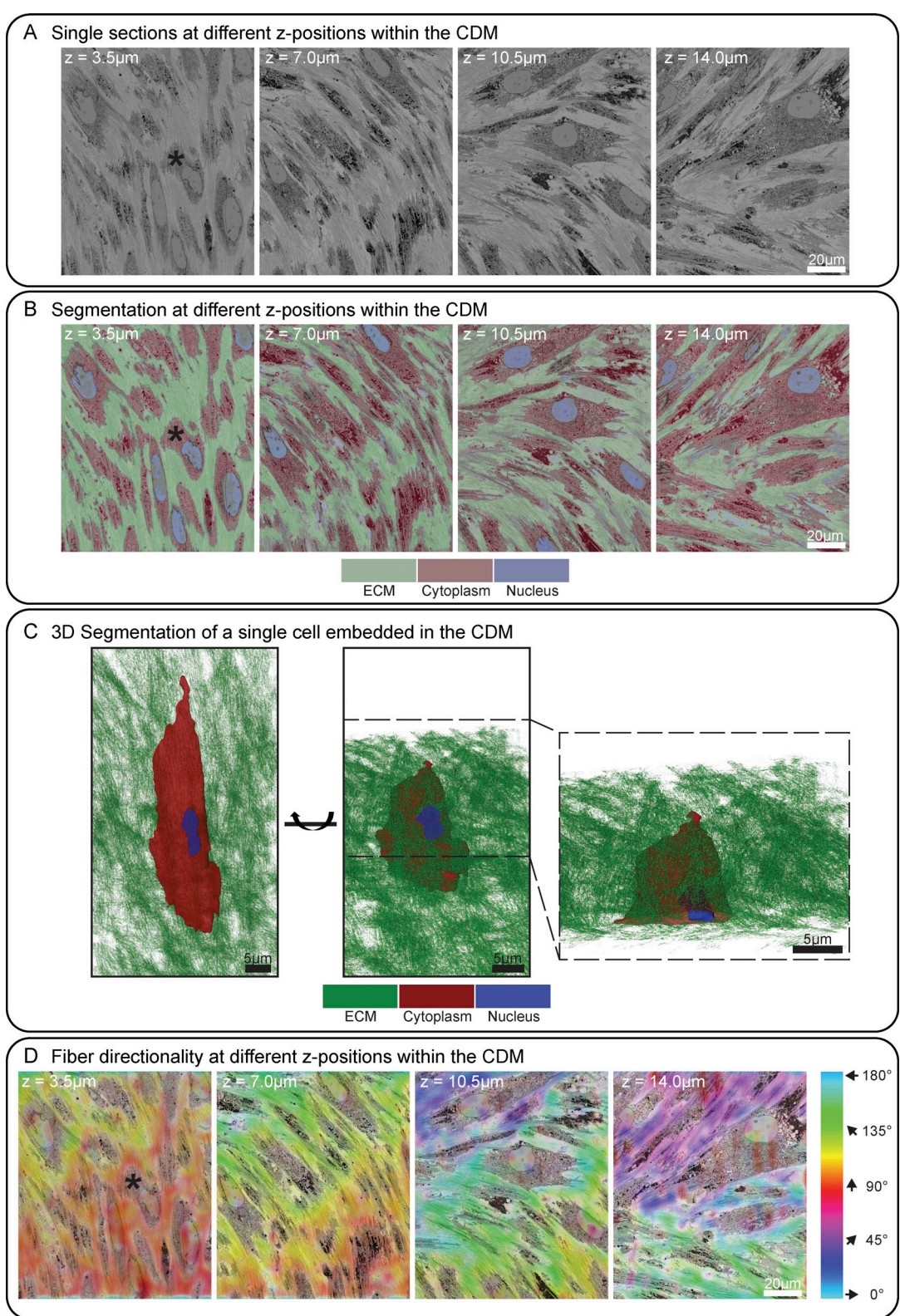

Figure S1.   **Room-temperature SEM array tomography of TIFF CDMs grown for 14 days. (A)** Four exemplary 70 nm thick sections from different z-height positions within the CDM are shown. **(B)** The same slices shown in A were segmented for nuclei, cytoplasm, and ECM using Ilastik (Berg et al., 2019) and are colored as indicated. **(C)** 3D segmentation of an exemplary cell embedded within the CDM (denoted by a black asterisk in A), shown in a top (left) and oblique view (middle). On the right side, an ortho-slice view of the same cell is shown, highlighting the thin z-height of the cell embedded within the ECM. Please note that the other cells in the vicinity of the segmented cell have been omitted from this view to facilitate visualization. Color code of the segmentation is indicated below. **(D)** The directionality of cells and fibers is color-coded as indicated on the right, corresponding to the angles of cells and fibers. The same z-positions as shown in A are depicted to show the change in directionality throughout the height of the CDM. Scale bar dimensions are annotated in the figure.

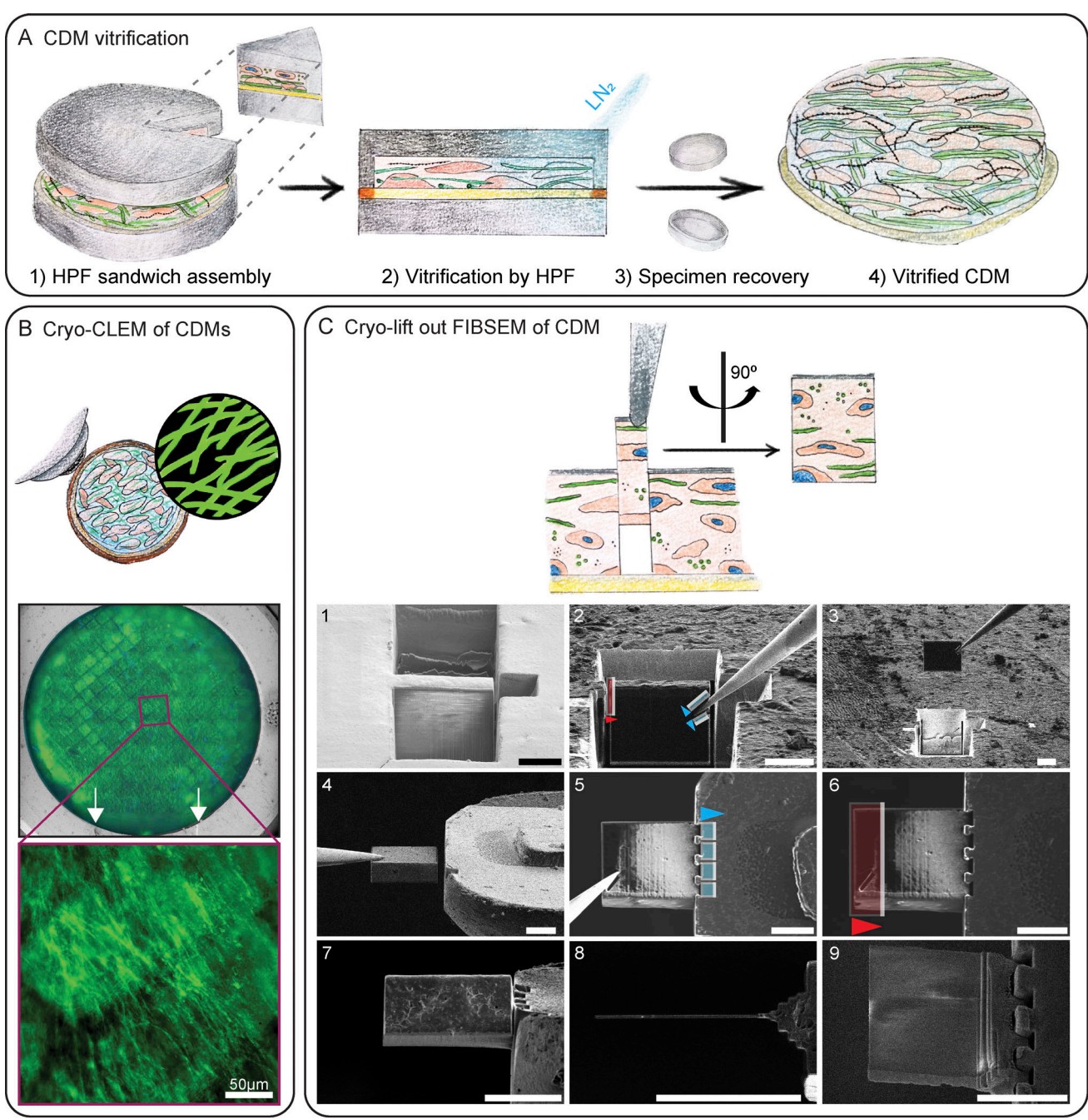

Figure S2.   **HPF and cryo-lamella preparation of CDMs for cryo-ET. (A)** Schematic workflow of on-grid CDM vitrification: (1) sandwich assembly of on-grid CDMs, live-stained for collagen; (2) vitrification via HPF; (3) specimen recovery of (4) vitrified CDM. **(B)** All specimens are screened for their quality and to define regions of interest (ROI, annotated with a purple rectangle) by cryofluorescence light microscopy. A magnified image of the ROI, showing collagen fibers is depicted below. The reflected light is used to define landmarks for correlation, such as the milling window on the FIBSEM Autogrid, indicated by white arrows. **(C)** Cryo-lift-out FIBSEM workflow. Trenches are milled to isolate the lift-out (1), which is attached to a micromanipulator needle by redeposition milling (blue patterns, 2) and prior to cutting it loose from the bulk sample (red pattern, 2). The lift-out is extracted from the bulk sample (3) and attached to a finger-like protrusion on a half-moon grid by redeposition milling (4–5, blue patterns). The needle is cleanly pulled off and its attachment site is removed from the lift-out by FIB milling (red pattern) (6). The lift-out can be milled into a thin lamella (7–9) compatible with cryo-ET. Arrowheads indicate milling direction. Scale bar dimensions are 10 µm unless annotated otherwise.

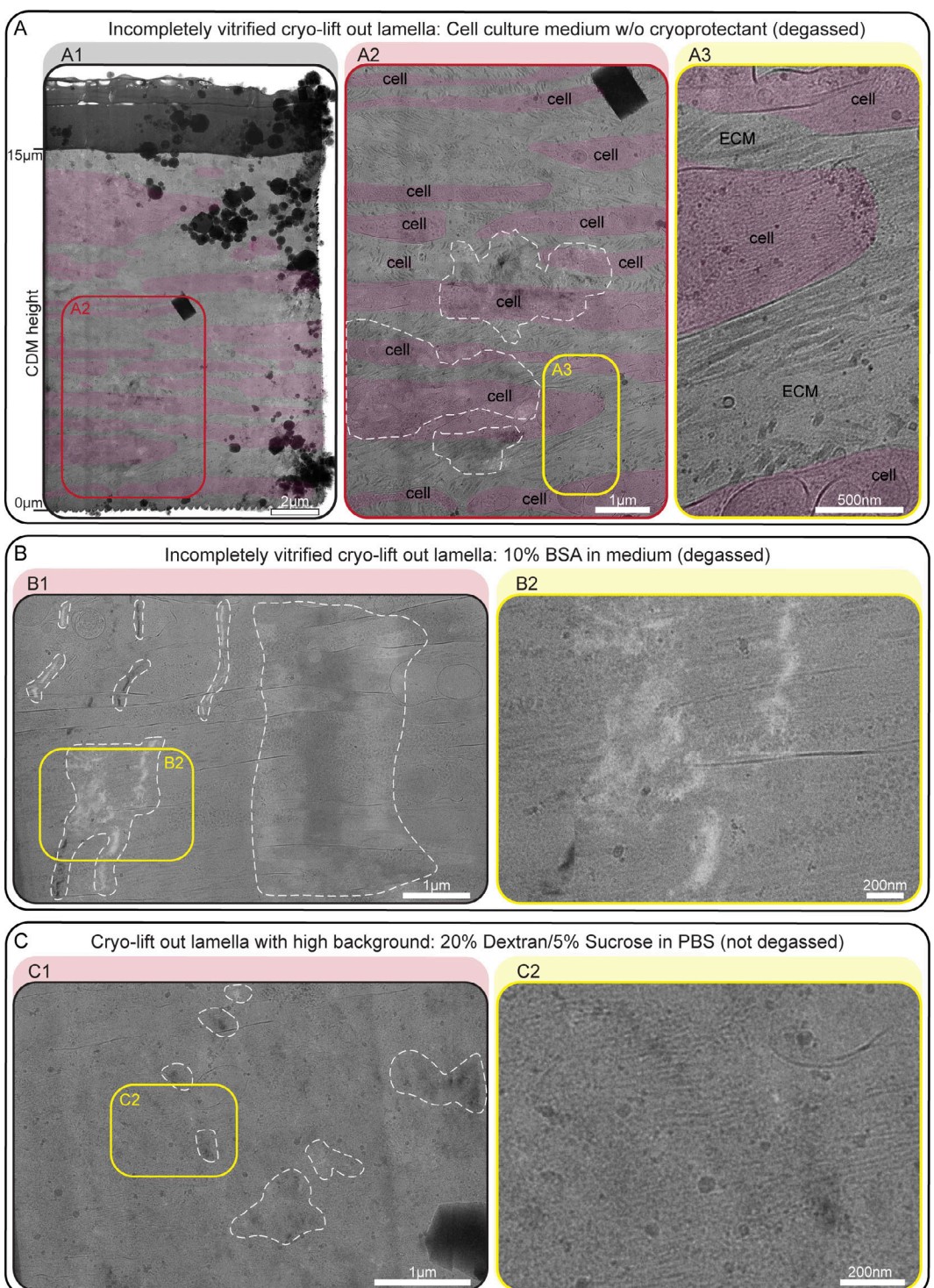

Figure S3. **Examples of incompletely vitrified CDM lift-out lamellae. (A)** CDMs were vitrified in cell culture medium without cryoprotectants. **(A1)** Overview of a whole cryo-lift-out lamella. The lamella covers roughly 15 µm of CDM depth, ranging from proximal to the EM grid substrate (z = 0 µm) to close to the CDM surface (z = 15 µm). Cell bodies are annotated in transparent red color. **(A2)** Zoom-in of the lamella as annotated with a red rectangle in A1. Areas with reflections caused by incomplete vitrification are marked with a white, dashed line. **(A3)** Zoom-in of the area annotated with a yellow rectangle in A2, showing high-contrast ECM structures of interest. **(B)** Example of a lamella with incomplete vitrification, despite the use of cryoprotectant (cell culture medium, with 10% BSA, degassed). **(B1)** Lamella overview. Strong reflections are observed throughout the area (marked with a white dashed line). **(B2)** Zoom-in into the lamella, as annotated by a yellow rectangle in B1. The reflections obscure cell and ECM details, while the background would have been acceptable. **(C)** Example of a lamella with too high background, introduced by the cryoprotection buffer (20% dextran, 5% sucrose in PBS, without degassing). **(C1)** Lamella overview. Weak reflections can be seen throughout the area (white dashed line), resulting in a categorization of the vitrification status as incomplete. **(C2)** Zoom-in into the lamella, annotated by a yellow rectangle in C1. The high background reduces the visibility of cellular and ECM structures. Scale bar dimensions are annotated in the figure.

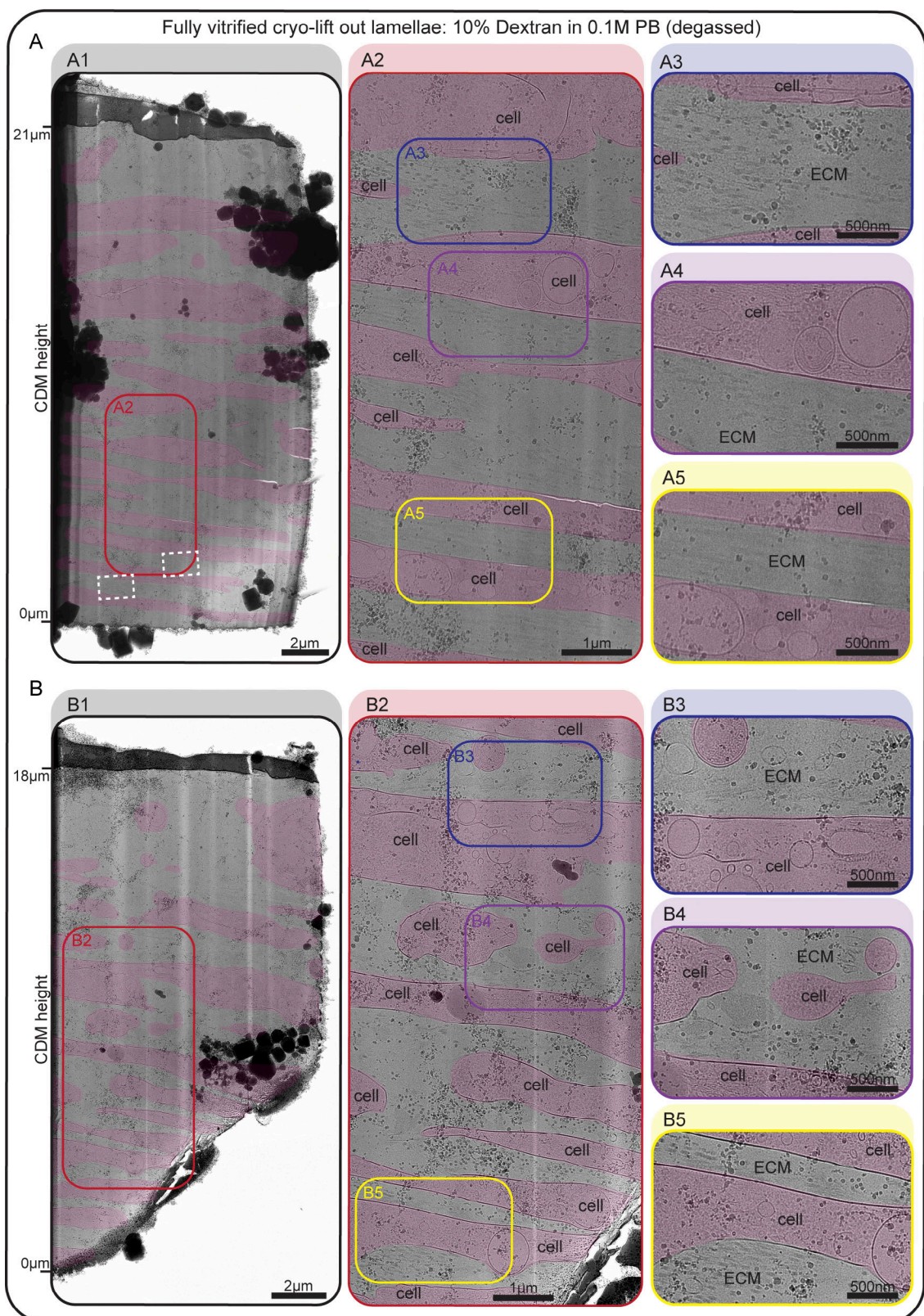

Figure S4. **Additional examples of completely vitrified cryo-lift-out lamellae. (A and B)** Two additional, completely vitrified lamellae of day 16 CDMs, high-pressure frozen in degassed 0.1 M PB with 10% dextran. White dashed rectangles denote the areas of acquisition for the tomograms shown in Fig. S5. An overview map of each whole lamella is shown in A1 and B1. The CDM height is indicated on the left, starting from 0 µm close to the EM grid and up to 21 and 18 µm at the surface of the CDM, respectively. Areas featuring structures of interest are depicted at higher magnification in A2 and B2. Three ROIs are highlighted in each example and shown at higher magnification on the right (A3–A5 and B3–B5). Cell areas are annotated with a transparent red color, ECM areas are labeled as such. Scale bar dimensions are annotated in the figure. The experimental conditions of the shown CDMs are identical to the data shown in Fig. 2.

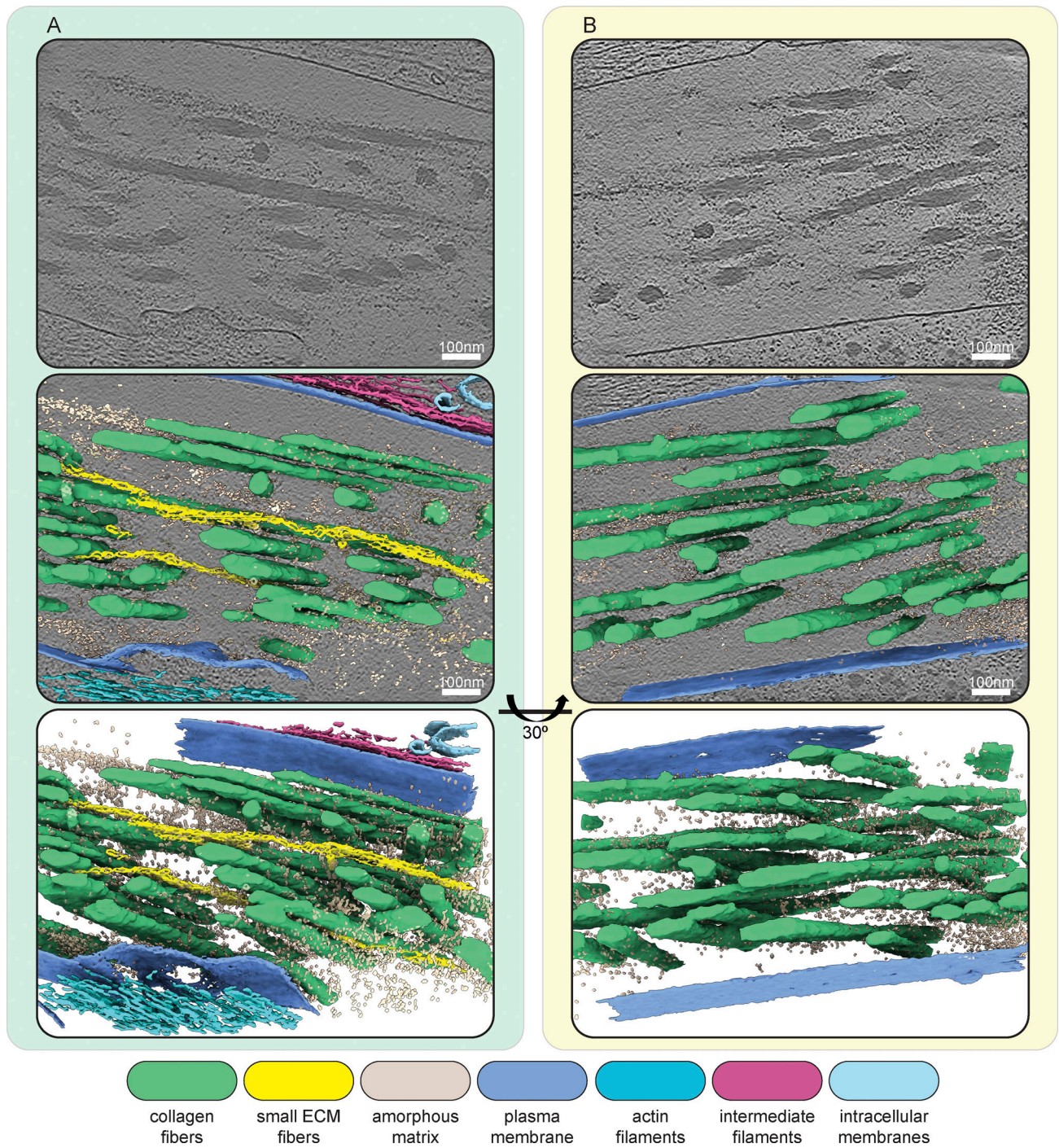

**Figure S5. Additional examples of segmented cryo-electron tomograms. (A and B)** Segmentations of two exemplary IsoNet-processed tomograms acquired on the cryo-lift-out lamella shown in Fig. S4 A. The top panels show a single central slice (thickness of 1.71 nm) of each respective tomogram. Cell and ECM fibers run in an oblique angle to the lamella, resulting in a side view of intra- and extracellular filaments. Middle panels show segmentations of tomograms overlaid over the tomogram slice. The bottom panels show an oblique view of just the segmentation volume. Scale bars indicate 100 nm. The color scheme for the different segmented cellular and ECM components is described in the figure.

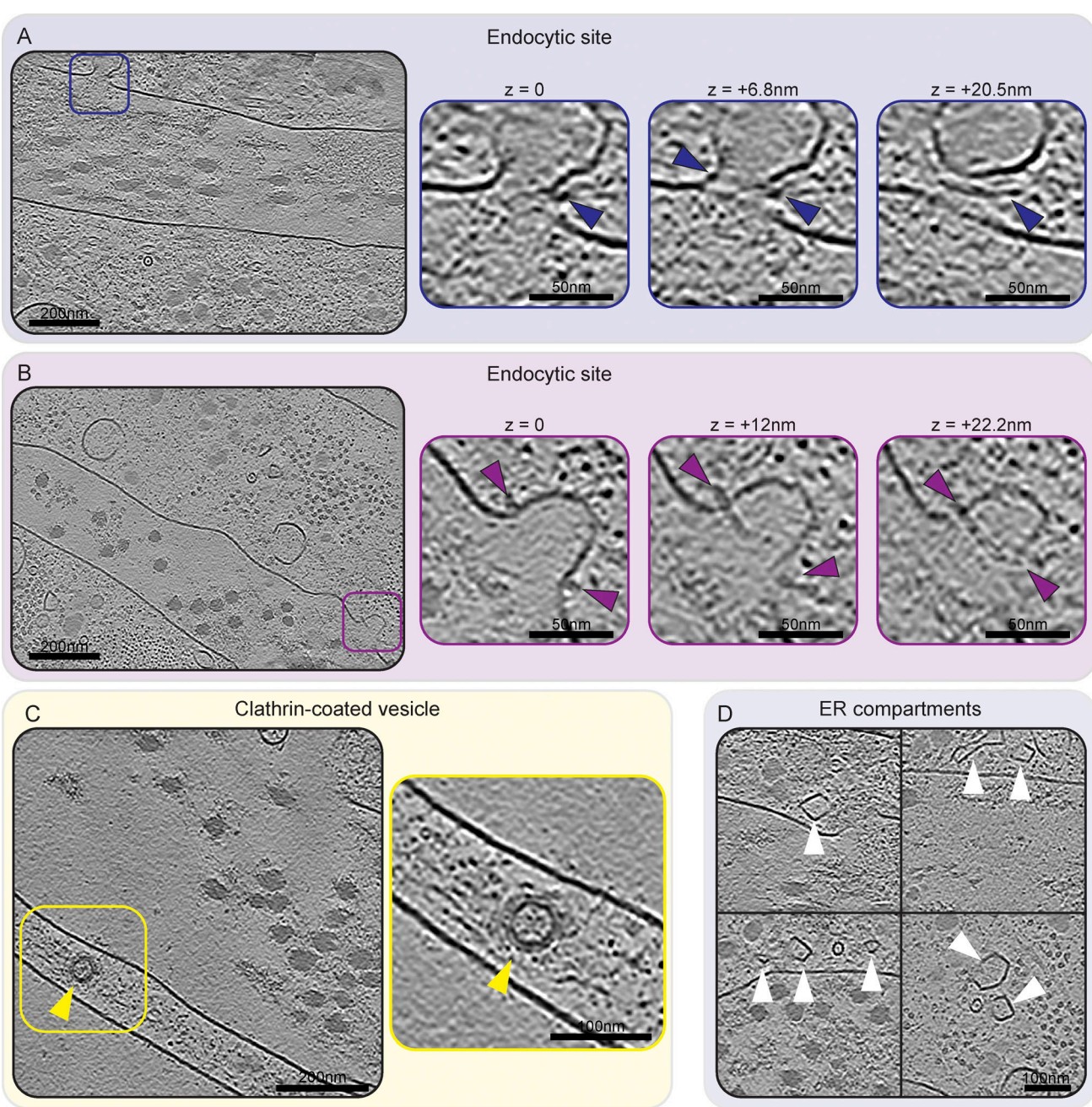

Figure S6. **Membrane compartments—endocytic sites, clathrin-coated vesicles, and ER compartments. (A and B)** Single slices of two IsoNet-processed tomograms are shown on the left. Both tomograms show an endocytic site, highlighted by blue (A) and purple (B) squares, respectively. For both tomograms, these sites are shown at higher magnification and different z-positions to better illustrate the structural details. Endocytosis-associated proteins, potentially BAR-domain proteins, can be seen assembled at the neck of the forming vesicle, indicated by blue and purple arrowheads. The difference in z-position is indicated above each image. **(C)** A single central slice of an IsoNet-processed tomogram shows a clathrin-coated vesicle annotated by a yellow arrowhead. **(D)** Four exemplary regions containing ER compartments (annotated by white arrowheads) in IsoNet-processed tomograms are shown. All tomogram slices have a thickness of 1.71 nm. Scale bar dimensions are annotated in the figure.

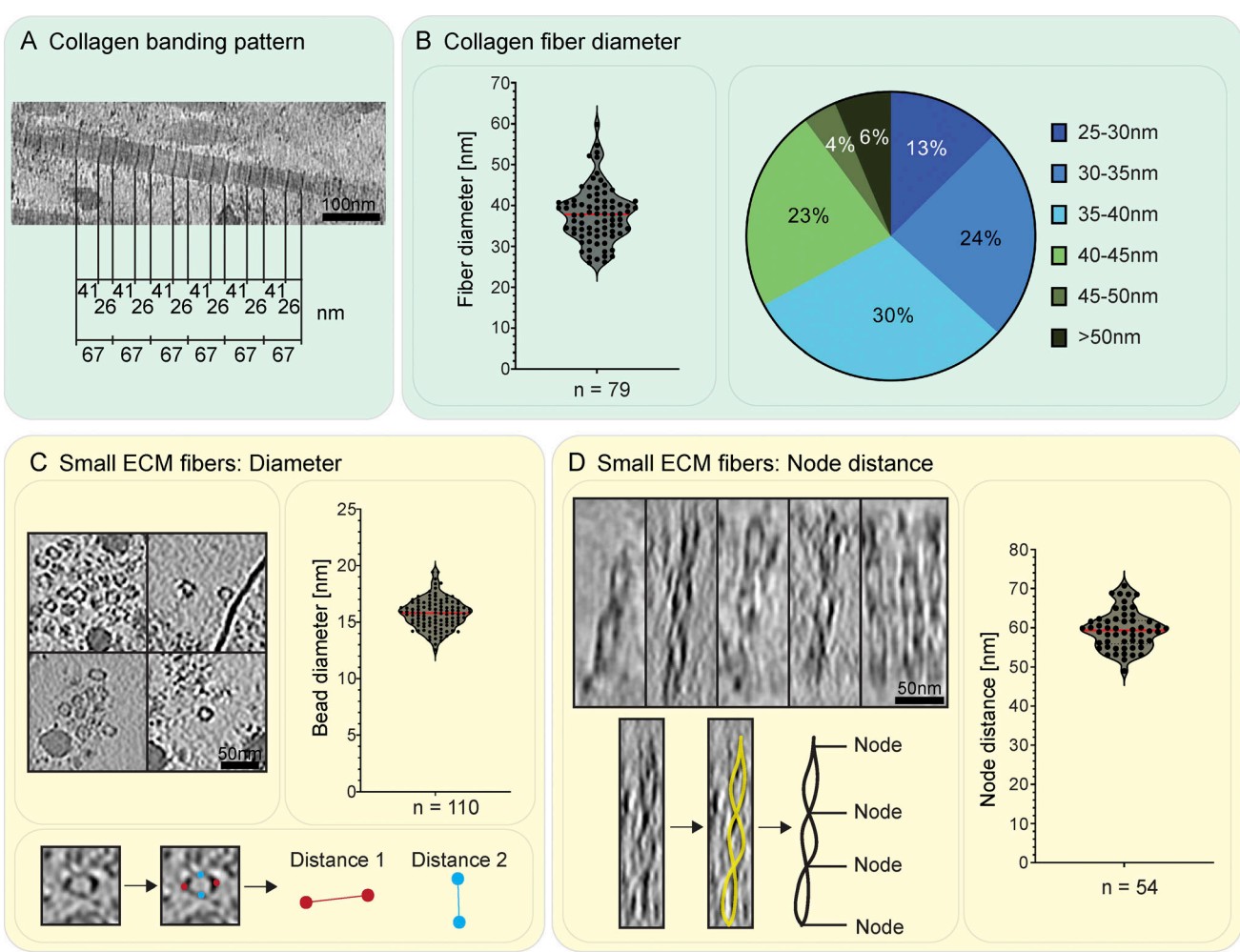

Figure S7.   **Details of ECM fibers. (A)** Focused view on the banding pattern of collagen fibers. The shown tomogram was reconstructed using a SIRT-like filter to enhance the visibility of the banding pattern. The typical collagen repeat pattern of a 41 nm wide band followed by a 26 nm narrow band, summing up to an overall repeat of 67 nm, is indicated. **(B)** Fiber diameter of 79 collagen fibers as measured in cross-section view (to reduce missing-wedge effect-caused inaccuracies). Each single measurement is shown as a dot in a violin plot. A red line indicates the median. On average, collagen fibers have a diameter of ∼38 nm (SD = 6.5 nm), ranging from 26 to 60 nm. The distribution in diameter of the measured fibers is shown in more detail in the pie-chart on the right. **(C and D)** Characterization of small ECM fibers by measuring their diameter (C) and node-to-node repeat distance (D) in IsoNet-corrected tomograms. **(C)** Four exemplary images showing one or more small ECM fibers (left). Their diameter has been measured at their widest point. As fibers were not round, the average of two distances, measured perpendicular to each other, was calculated. 110 individual measurements are displayed in the respective violin plot shown on the right, with the red line indicating the median. The average bead diameter is 15.74 nm (SD = ±1.31). **(D)** The node-to-node distance of the small ECM fibers was determined for 54 individual measurements. Exemplary images of IsoNet-corrected tomogram regions used for measurements are shown on the left. Below, one exemplary filament is traced in yellow, with the tracing shown as standalone in black next to it. The measurements are summarized in the violin plot on the right, the red line indicates the median. Overall, the node distance is at average 60.30 nm (SD = ±7.82 nm). Scale bar dimensions are annotated within the figure.

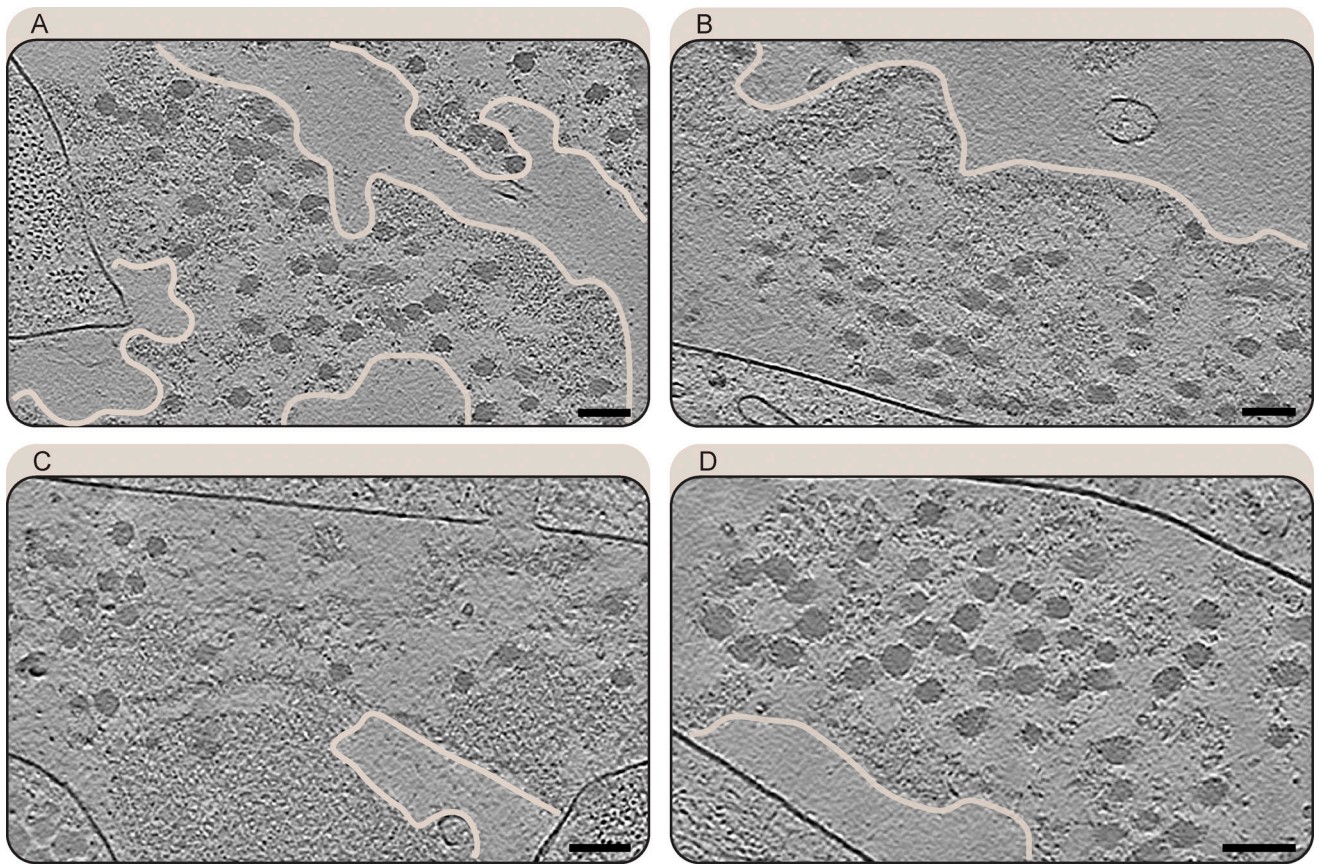

Figure S8. **Amorphous matrix details. (A–D)** Slices (1.71 nm thickness) of IsoNet-processed tomograms showing the amorphous matrix co-localizing with collagen fibers. Sharp edges (traced in light peach for better visibility) delineate the amorphous matrix from empty extracellular space. Plasma membranes surrounding cells appear as strong black lines. Scale bar dimensions are 100 nm.

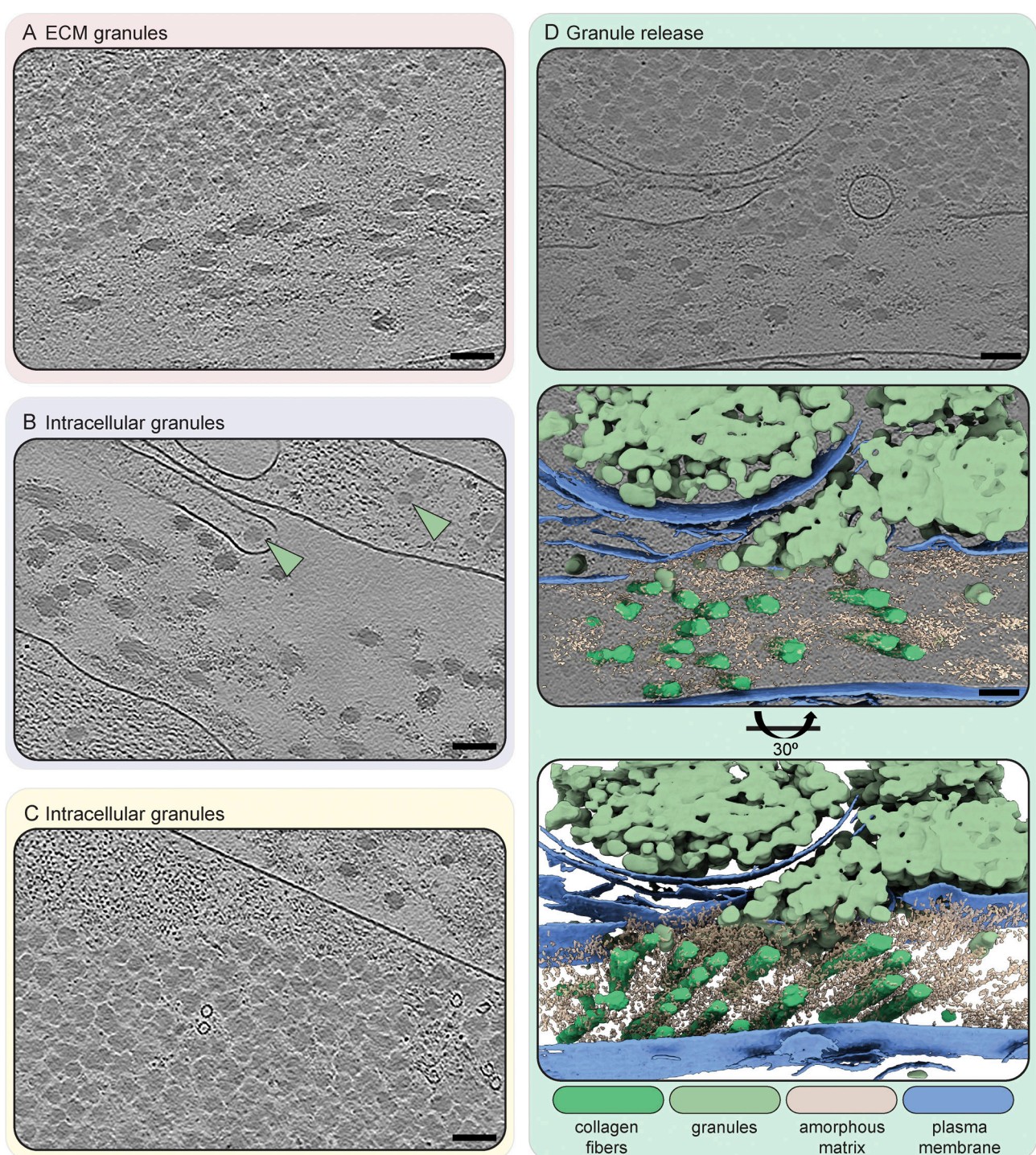

Figure S9. **Extra- and intracellular ECM-associated granules. (A)** Granules in the extracellular space close to collagen fibers and amorphous matrix. **(B)** Granules in the intracellular space, in the main cell body as well as in a cellular protrusion. Granules are indicated by a light green arrowhead. **(C)** Intracellular granules close to the plasma membrane, next to actin filaments and with two microtubules interspersed. **(D)** A granule release site, where granules are in both the intra- and extracellular space. The plasma membrane is partially interrupted. The top panel shows a slice of a tomogram acquired at this site. This tomogram was segmented using deep-learning based software (Dragonfly) and visualized using ChimeraX. The middle panel shows an overlay of the segmentation and the tomogram panel shown above. The lower panel shows the segmentation at an oblique angle of 30° for better visibility of the ECM fibers. The color scheme for the different segmented cellular and ECM components is described in the figure. All slices (1.71 nm thickness) are from IsoNet-processed tomograms. Scale bar dimensions are 100 nm.

Video 1.   **Movie of the tomogram and segmentation shown in** Fig. 3 A**.** Movie of the tomogram shown in Fig. 3 A with and without segmentation of intra- and extracellular structures in TIFF-derived CDM. Coloring of the segmentations is as described in the legend of Fig. 3 A.

Video 2.   **Movie of the tomogram and segmentation shown in** Fig. 3 B**.** Movie of the tomogram shown in Fig. 3 B with and without segmentation of intra- and extracellular structures in TIFF-derived CDM. Coloring of the segmentations is as described in the legend of Fig. 3 B.

Video 3.   **Movie of the tomogram and segmentation show in** Fig. S9 D**.** Movie of the tomogram shown in Fig. S9 D with and without segmentation of intra- and extracellular structures in TIFF-derived CDM. Coloring of the segmentations is as described in the legend of Fig. S9 D.

Video 4.   **Movie showing features following a linear filament-like trajectory.** The movie displays a tomogram featuring an unidentified linear filament-like trajectory in TIFF-derived CDM. The filament-like structure is annotated with a black arrowhead. The small bead-like ECM filaments are annotated with yellow arrowheads or a yellow ellipsoid.

Video 5.   **Movie showing features following a linear filament-like trajectory.** The movie displays a second example of a tomogram featuring an unidentified linear filament-like trajectory in TIFF-derived CDM. The filament-like structure is annotated with a black arrowhead. The small bead-like ECM filaments are annotated with yellow arrowheads or a yellow ellipsoid.

**Provided online are three tables. Table S1 shows examples of reported dimensions for different ECM fibers. Table S2 shows predominant ECM proteins in TIFF CDMs identified by mass spectrometry. Table S3 shows scouting of different cryoprotectants and buffers for their vitrification potential.**

