## [Peer Review File · The Journal of Cell Biology]

Lift-out cryo-FIBSEM and cryo-ET reveal the ultrastructural landscape of extracellular matrix

Bettina Zens, Florian Faessler, Jesse Hansen, Robert Hauschild, Julia Datler, Victor-Valentin Hodirna, Vanessa Zheden, Jonna Alanko, Michael Sixt, and Florian Schur

Corresponding Author(s): Florian Schur, Institute of Science and Technology Austria

Review Timeline:

Submission Date:	2023-09-26
Editorial Decision:	2023-11-07
Revision Received:	2024-02-19
Editorial Decision:	2024-02-23
Revision Received:	2024-02-28

Monitoring Editor: Kenneth Yamada

Scientific Editor: Dan Simon

Transaction Report:

DOI: <https://doi.org/10.1083/jcb.202309125>

November 7, 2023

Re: JCB manuscript #202309125

Dr. Florian KM Schur
Institute of Science and Technology Austria
Am Campus 1
Klosterneuburg 3400
Austria

Dear Dr. Schur,

Thank you for submitting your manuscript entitled "Unveiling the ultrastructural landscape of extracellular matrix via lift-out cryo-FIBSEM and cryo-ET" to the Journal of Cell Biology. It has been evaluated by three acknowledged leaders in fields spanning the various elements of your manuscript. As you can see from the appended reviews, they are positive about the value of your cryo-FIBSEM and -ET approaches. They also raise a variety of points that will need to be addressed.

The relatively minor concerns and useful suggestions are numerous, but all appear to be readily addressable. Besides resolving the technical questions and wording/citation concerns, it would indeed be helpful and reassuring if possible to confirm that the filaments are indeed collagen fibers using second harmonic generation, though this approach is not absolutely essential.

We look forward to receiving a revised manuscript from you with a point-by-point response to each of the comments of these conscientious reviewers. We will re-review the manuscript at the Editor level to confirm that each of the points have been resolved within reason. We thank you for your interest in the Journal of Cell Biology and for this valuable contribution.

GENERAL GUIDELINES:

Text limits: Character count for a Report is < 20,000, not including spaces. Count includes title page, abstract, introduction, the joint Results & Discussion, and acknowledgments. Count does not include materials and methods, figure legends, references, tables, or supplemental legends.

Figures: Reports may have up to 5 main text figures. To avoid delays in production, figures must be prepared according to the policies outlined in our Instructions to Authors, under Data Presentation, <https://jcb.rupress.org/site/misc/ifora.xhtml>. All figures in accepted manuscripts will be screened prior to publication.

*****IMPORTANT:** It is JCB policy that if requested, original data images must be made available. Failure to provide original images upon request will result in unavoidable delays in publication. Please ensure that you have access to all original microscopy and blot data images before submitting your revision. ***

Supplemental information: There are strict limits on the allowable amount of supplemental data. Reports may have up to 3 supplemental figures. Up to 10 supplemental videos or flash animations are allowed. A summary of all supplemental material should appear at the end of the Materials and methods section.

Please note that JCB now requires authors to submit Source Data used to generate figures containing gels and Western blots with all revised manuscripts. This Source Data consists of fully uncropped and unprocessed images for each gel/blot displayed in the main and supplemental figures. If your paper will include cropped gel and/or blot images, please be sure to provide one Source Data file for each figure that contains gels and/or blots along with your revised manuscript files. File names for Source Data figures should be alphanumeric without any spaces or special characters (i.e., SourceDataF#, where F# refers to the associated main figure number or SourceDataFS# for those associated with Supplementary figures). The lanes of the gels/blots should be labeled as they are in the associated figure, the place where cropping was applied should be marked (with a box), and molecular weight/size standards should be labeled wherever possible. Source Data files will be made available to reviewers during evaluation of revised manuscripts and, if your paper is eventually published in JCB, the files will be directly linked to specific figures in the published article.

The typical timeframe for revisions is three to four months. While most universities and institutes have reopened labs and

allowed researchers to begin working at nearly pre-pandemic levels, we at JCB realize that the lingering effects of the COVID-19 pandemic may still be impacting some aspects of your work, including the acquisition of equipment and reagents. Therefore, if you anticipate any difficulties in meeting this aforementioned revision time limit, please contact us and we can work with you to find an appropriate time frame for resubmission. Please note that papers are generally considered through only one revision cycle, so any revised manuscript will likely be either accepted or rejected.

Thank you for this interesting contribution to Journal of Cell Biology. You can contact us at the journal office with any questions at cellbio@rockefeller.edu.

With kind regards,

Kenneth M. Yamada, MD, PhD
Senior Editor
Journal of Cell Biology

Dan Simon, PhD
Scientific Editor
Journal of Cell Biology

Reviewer #1 (Comments to the Authors (Required)):

I congratulate the authors on this heroic effort aiming to provide a first structural 3D atlas of the ECM at molecular resolution. The systematic effort to optimize specimen preparation of cell culture-based 3D-ECM platforms for compatibility with cryo-FIB lift-out and high quality cryo-ET provide a valuable starting point for studies focused on complex cell-ECM interactions that better recapitulate in vivo scenarios in a controlled 3D in vitro system. The quality of the data is outstanding and the accompanying annotations are stunning. The authors complement their structural characterization with systematic proteomics and light microscopy-based measurements. While the manuscript remains descriptive, the work in my view opens an exciting avenue for the study cell-matrix interfaces and illuminating critical gaps in our knowledge of ECM synthesis, secretion and assembly, and the effect of critical mutations.

I recommend publication, and would have liked to see more quantification of the volume fractions of the different ECM structures identified. These might be useful in the future, especially if MS data is presented in parallel in terms of protein abundance, to possibly identify potential candidates for newly identified ECM structures (more below).

Minor comments:

The manuscript text is at times too wordy and therefore convoluted, especially in the introduction and in some of the results description. It may benefit from simplification and consolidation.

Abstract: "sample thinning by cryo-lift out focused ion beam milling". Lift out is an extraction procedure. Microfabrication and thinning are done by FIB. I recommend rephrasing for clarity to a broader audience.
"This allows us to visualize ECM for the first time in its hydrated, cellular context". The authors may want to reward "cellular" to "native" or similar.

Figure 1: I wonder whether the authors could comment on the origin of the observed directionality of both the extracellular and intracellular filaments. There is more information in the supplementary figures, but it is not explained or discussed. Does this evolve over time from cell seeding to the end point of the cryo-EM investigation? Is it directed by something in the EM grid substrate?

Page 3: "Altogether, these results confirmed that CDMs harvested on or later than Day 14 represent bona fide ECM assemblies". It would be useful if the authors could point to previous publications on the expected morphology of "bona fide" ECM in support of their conclusion.

Page 4: with regard to the testing of the cryo-protectants, it would be useful if the authors could provide further information, and possibly references. For example, for Dextran, it is important to provide the molecular weight as this is an important factor for osmolarity and discuss evaluation of the effect of osmolarity on cell/extracellular native preservation in this context.

Page 4: how do the authors interpret the "empty areas" in the extracellular space? This is where a parallel presentation of the ECM proteomics could be useful.

Figure 2: how do the authors interpret the large vacuoles observed primarily in cells located towards the top of their lamellae? The authors may want to discuss this if appears to be a common feature.

Page 6: "The variation in both diameter and repeat pattern along the filament z-axis suggests extensibility and assembly variability." And "the amorphous matrix was observed only in close proximity to ECM fibers... This suggests a co-assembly between the components in these entities." These are unnecessary overinterpretations of the data. I recommend that the authors rephrase, especially when the structures cannot be identified.

Page 6, sections "The absence of other filament-like structures in CDMs": here it would be useful to explicitly spell out the candidates from proteomics measurements, or the previous reports the authors mention (without providing references!) and the expected structural features, potentially even from structure predictions. For example, Table 2 "ECM proteins in TIFF CDMs identified by MS" could perhaps be presented in a plot form that also reflects relative abundance (possibly only the fibril-forming proteins, while others are aggregated, for simplicity) in the main text and related to the related volume fractions occupied by the different ECM structures as observed by cryo-ET.

I recommend that the authors include in their discussion/conclusion some mention on the potential of "waffle" or "Serial lif-out" methodologies to gain higher throughput and more insights into similar model systems given the dimensions of their specimens.

Movies S4 and S4: some of the fuzzy amorphous density appears to be continuous with the Col-I fibrils. It might be useful for the authors to mention this if this appears to be a common feature in other data as it might indicate that these structures are precursors of Col-I assembly.

Figure S1: scale bar size missing in b.

Methods, cryo-FIB: unclear at which point the half-moon grid are mounted into autogrids. Please add.

Data availability: it would be extremely valuable for the community if the authors deposit the tomograms and segmentations associated with this work. I would even strongly recommend deposition of the full data, from raw movies to segmentations in EMPIAR.

Reviewer #2 (Comments to the Authors (Required)):

In this article entitled "Unveiling the ultrastructural landscape of extracellular matrix via lift-out cryo-FIBSEM and cryo-ET", Zens et al. reporting a technical development based on cryoEM and FIBSEM technologies. They adapted and combined methodologies (references 24, 28, 31 and 36) to enable visualization of cell derived matrices in a hydrated context. This is of outmost relevance considering the deleterious effects of classic protocols including chemical fixation and dehydration steps. This article is well written and well-illustrated. It has an appreciable part of "trial and error" reporting of the scientific process, and results in establishment of a successful and promising protocol for the study and for future studies. This is the primary merit of the study and as such may become seminal in the field.

Limitations of the manuscript are related to the relatively low number of conclusive points that are made by the authors. Indeed, the use of a single biological condition (TIFF derived CDM) is somewhat limiting the general value of the study. Having said that, owing to the laborious nature of this approach, limiting efforts to a single sample time is understandable.

Major comments:

The authors should consider toning down this statement:

"Another aggravating factor for the structural annotation of ECMs is the heterogeneity of tissue-derived ECM material. In contrast, cell derived matrices (CDMs) are a highly adaptable and versatile tool that is increasingly used to recapitulate the complexity of native tissue ECM (29-31)."

In reality, CMD are not recapitulating the complexity of native tissue, which is the results of many cell types interacting together and jointly contributing to the matrix compositions and remodelling. Only in vivo observation can fully recapitulate in vivo complexity and CDM is rather a (valuable) reductionist in vitro model system to study matrix biology.

The authors are taking care not to overinterpret their findings and this is commendable. However, it would be beneficial for the reader to if some more concrete conclusions could be drawn.
some conclusions out of this work.

Related to this topic, the "beaded filaments" and the absence of fibronectin visualization are surprising and need more explanation.

Can the authors rule out the possibility that "amorphous ECM" is in fact not well-preserved during sample preparation?

Can the rare fibers that are observed in the amorphous ECM be a low percentage of preserved material?
Can this be fibronectin filaments?
Can the beaded filaments be fibronectin filaments?

One major limitation of the study is the lack of comparisons between for example condition or treatments that are expected to influence the CDM. For example, genetic manipulation of the TIFF or CDM made by another cell type. If these are beyond the scope of this study, the authors should discuss this point and mention this limitation.

Minor comments:

The authors abstract first paragraph is somewhat misleading, especially when mentioning Col VI assembly as remaining "enigmatic", considering that there is no conclusive statement made in the manuscript regarding Col VI.

Fig S1 : Please include original images of panel A, before annotation.

Fig S6 : "enodcytosis" should be corrected to "endocytosis".

Fig S9 : If available, please show low magnification pictures for panel D. The same way it has been done in Fig S6 A, for instance.

These expressions/sentences are unclear or inappropriate to my opinion.

"There is no unambiguous consensus..."

"molecular sociology..."

Reviewer #3 (Comments to the Authors (Required)):

This study is an exquisite example of how new technology can facilitate novel information. The only areas of potential improvement are some missing citations to credit the original CDM work and methodology (and similar) as well as a suggestion to combine the Cryo-generated data with SHG (second harmonic generation of polarized light) using the same 3D sample. The latter is merely a suggestion that could significantly improve the interpretation of the data (regarding collagen fibers).

Specific MINOR suggestions:

1. In the introduction, when CDMs are first described as a valid in vivo mimicry approach, the authors may want to cite the original paper and methods depicting these cell/ECM functional units:

a. Paper: doi: 10.1126/science.1064829

b. Methods:

i. Original 2002: <https://doi.org/10.1002/0471143030.cb1009s16>

ii. Updated in 2007: <https://doi.org/10.1002/0471143030.cb1009s33>

iii. Updated again in 2016: PMID: PMC5058441

2. A few sentences later in the introduction, when highlighting some "fundamental research" examples, the authors should include cancer research and cite some of the Cukierman team's work:

a. *elife* 2017

b. *Matrix Biology* 2019

c. *Communications Biology* 2020

d. *Cancer Discovery* 2021

e. *CRC* 2022

3. Technically, in results shown in Figure 1C, the day 14 fibronectin staining looks as if the immunofluorescence followed a process whereby fixing of the cells was done first and permeabilization followed. The original methods have provided troubleshooting for this process (to avoid the apparent void in fibers) by conducting a simultaneous fix/permeabilization step followed by added fixing (published in this version of CDM production methods and quality control analyses: PMID: PMC5058441). The authors may want to repeat this using this advice and potentially generate a higher quality image/figure.

4. Regarding data in supplemental figure 2, CDMs have been shown to mature enough (e.g., incorporate enough fibrous collagen) in vitro to include collagen fibers that are detected using the second harmonic generation of polarized light (obtained with most multiphoton microscopes: PMID: PMC7442735). Hence, the authors may want to include this orthogonal approach to support their interpretation of the data (to support that the 25-60 nm filaments are indeed collagen fibers as suggested) and demonstrate the high quality of their CDMs.

5. The statement suggesting that the growth of new CDM (fibers) preferentially takes place on the top layers is not well justified

and should probably be omitted, while the density of fibers statement could prevail.

Response to Reviewers:

We thank the reviewers for their evaluation of our manuscript and have responded to their questions point-by-point below.

Reviewer #1 (Comments to the Authors (Required)):

I congratulate the authors on this heroic effort aiming to provide a first structural 3D atlas of the ECM at molecular resolution. The systematic effort to optimize specimen preparation of cell culture-based 3D-ECM platforms for compatibility with cryo-FIB lift-out and high quality cryo-ET provide a valuable starting point for studies focused on complex cell-ECM interactions that better recapitulate in vivo scenarios in a controlled 3D in vitro system. The quality of the data is outstanding and the accompanying annotations are stunning. The authors complement their structural characterization with systematic proteomics and light microscopy-based measurements. While the manuscript remains descriptive, the work in my view opens an exciting avenue for the study cell-matrix interfaces and illuminating critical gaps in our knowledge of ECM synthesis, secretion and assembly, and the effect of critical mutations.

We thank the reviewer for the positive comments.

I recommend publication, and would have liked to see more quantification of the volume fractions of the different ECM structures identified. These might be useful in the future, especially if MS data is presented in parallel in terms of protein abundance, to possibly identify potential candidates for newly identified ECM structures (more below).

We have performed the suggested quantifications and provide more details on the results and the limitations associated with such quantifications below.

Minor comments:

The manuscript text is at times too wordy and therefore convoluted, especially in the introduction and in some of the results description. It may benefit from simplification and consolidation.

In the course of revision, the text has changed in several instances. We hope this has consolidated the text.

Abstract: "sample thinning by cryo-lift out focused ion beam milling". Lift out is an extraction procedure. Microfabrication and thinning are done by FIB. I recommend rephrasing for clarity to a broader audience.

"This allows us to visualize ECM for the first time in its hydrated, cellular context". The authors may want to reward "cellular" to "native" or similar.

We have reworded the sentences accordingly (changes in bold):

*"We have developed a 3D-ECM platform compatible with sample thinning **by cryo-focused ion beam milling, the lift-out extraction procedure and cryo-electron tomography.**"*

*“This allows us to visualize ECM for the first time in its hydrated, **native** context.”*

Figure 1: I wonder whether the authors could comment on the origin of the observed directionality of both the extracellular and intracellular filaments. There is more information in the supplementary figures, but it is not explained or discussed. Does this evolve over time from cell seeding to the end point of the cryo-EM investigation? Is it directed by something in the EM grid substrate?

Previous publications have shown the direct influence of cells, specifically fibroblasts, on the orientation of ECM fibers. This is mediated by cytoskeleton-ECM interactions such as fibrillar adhesions (Geiger et al, 2001; PMID: 11715046; Harris et al 1981, PMID: 7207616). Through these adhesions, cells have been reported to exert forces on ECM fibers such as fibronectin and collagens, and to align them (Piotrowski-Daspit et al., 2017; PMID: 28793224).

In agreement with this, throughout the 14+ days of CDM culturing, the fibroblasts seem to organize the ECM fibers as they are producing them (see Figure 1, where the alignment of cells and ECM is visible already early on). Accordingly, CDMs generated from TIFF cells and other epithelial cells have been shown to have highly aligned ECM fibers (Kaukonen et al., 2017, PMID: 29048422; Lansky et al., 2019, PMID: 32055794), matching our observations.

The EM substrate itself seemingly has no influence on this directionality, as the same organization can be observed when growing cells on glass cover slips or cell culture dishes (for example as shown in Figure S1) or as reported in Kaukonen et al., 2017.

We now also address this point more explicitly in the text:

“Previous publications have reported the direct influence of cells, specifically fibroblasts, on the orientation of ECM fibers, which are mediated by cytoskeleton-ECM interactions via adhesion complexes (Geiger et al., 2001; Harris et al., 1981). Cells align ECM fibers such as fibronectin and collagens by exerting forces through cell-ECM adhesion interactions (Piotrowski-Daspit et al., 2017).”

Page 3: "Altogether, these results confirmed that CDMs harvested on or later than Day 14 represent bona fide ECM assemblies". It would be useful if the authors could point to previous publications on the expected morphology of "bona fide" ECM in support of their conclusion.

We have now rephrased this sentence to further clarify the point made and added citations to support our statement. Specifically, we toned down our statement from *“representing bona fide ECM assemblies”* to *“mimicking ECM assemblies found in tissue”*.

(Changes in bold)

*“Altogether, these results confirmed that CDMs harvested on or later than Day 14 **mimic ECM assemblies found in tissue** (Fitzpatrick and McDevitt, 2015; Ahlfors and Billiar, 2007) and **should allow visualization of ECM components in their native environment of matrix-secreting cells.**”*

We are convinced that the composition of our CDMs is similar to ECM in tissue based on the MS data we provide, as well as our microscopy results which show higher-order assembly of ECM

components such as collagen and fibronectin. Previous publications use CDMs to study complex cellular behaviours such as cell migration (Hakkinen et al., 2011, PMID: 20929283), and in regenerative medicine (Fitzpatrick et al., 2015, PMID: 25530850; Bello et al., 2012, PMID: 11721649). While CDMs cannot represent an exact copy of ECM in tissues, they are a physiologically close representative system enabling us to study basic structures such as the ones described in this publication.

Page 4: with regard to the testing of the cryo-protectants, it would be useful if the authors could provide further information, and possibly references. For example, for Dextran, it is important to provide the molecular weight as this is an important factor for osmolarity and discuss evaluation of the effect of osmolarity on cell/extracellular native preservation in this context.

All cryo-protectants were either prepared in PBS, PB or cell medium. These buffers are commonly used in cell culture and tissue experiments by us and others, hence their physiological properties are evident.

Each cryo-protectant listed in Table S3 was chosen based on its previously described use for high pressure freezing. References for studies and protocols employing them were already given in the main text, as for example below.

“Many different cryoprotectant conditions commonly used in other experimental settings (49–54) resulted in insufficient vitrification in all tries (Figure S3B).”

For example, PVP is used in preparing brain tissue/neuron tissue, as shown in Borges-Merjane et al, 2020 (PMID: 31928842).

Similarly, BSA has been used by Tsang et al., 2018 (PMID: 29749931).

The cryo-protectant working best for us was Dextran (molecular weight of 40kDa). This Dextran has been used in previous studies performing high-pressure freezing such as Bharat et al, 2018 (PMID: 29681471).

In order to describe this in more detail in the revised manuscript, we now provide this information in the main text and methods (changes in bold).

*“A degassed cryoprotectant solution containing 10% **(w/v) high molecular weight Dextran (40 kDa)** in 0.1 M phosphate buffer (PB) showed the highest success and resulted in complete vitrification in several samples with low additional background (**Figure 2, Figure S4**). **High MW Dextran is a polymer that is non-penetrating and has been shown to have little osmotic effect on tissues (Dahl and Staehelin, 1989; Al-Amoudi, 2004), resulting in its routine use to facilitate vitrification of biological samples by HPF (Dahl and Staehelin, 1989; Sader et al., 2009; Bharat et al., 2018; Mesman, 2013; Zhang et al., 2021).**”*

We also added information to the legend of Table S3 as follows:

*“An overview table showing the tested cryoprotectant/buffer combinations, detailing the achieved vitrification status and the introduced background for each combination. **The buffers were chosen based on their physiological properties and their common use for biological specimens.**”*

Page 4: how do the authors interpret the “empty areas” in the extracellular space? This is where a parallel presentation of the ECM proteomics could be useful.

The reviewer makes a valuable point here. Indeed, we did not aim to imply that these areas are absolutely void of any molecules, but are devoid of large structures like collagen, amorphous density, or similar.

There are likely soluble proteins and/or sugars present in these areas, which we simply cannot visualize sufficiently with the used methodology. However, the “empty areas” could also match with being buffer/water-rich areas, fitting with a hydrogel.

Our HPF carriers were specifically designed to not compress the specimen and to retain the CDM in an unaltered state. It would be tempting to speculate that in tissues the ECM could be more compressed than is the case for our CDMs. Hence, it is conceivable that in a tissue, these spaces might be smaller. However, future work will be required to unambiguously show this.

While our proteomics data tells us that we have many secreted soluble, smaller proteins in the extracellular regions of our CDMs, such as growth factors, or proteases, it cannot give us any indication about the positioning of these proteins. Hence, we cannot confidently define which ECM components might be located in these seemingly empty spaces (or if any are).

Figure 2: how do the authors interpret the large vacuoles observed primarily in cells located towards the top of their lamellae? The authors may want to discuss this if appears to be a common feature.

The reviewer is correct, the presence of vesicles located in cells primarily towards the top of the CDM is interesting.

One potential interpretation of these large vesicles is that the top cells might be more active in the production and secretion of ECM components and the overall remodelling of the ECM. However, the content of these vesicles is unclear at this point. In addition, following the suggestion from reviewer #3, that our initial statement linking the growth of new CDM to the top layers should probably be omitted, we have refrained from further mentioning these vesicles in the manuscript.

Page 6: "The variation in both diameter and repeat pattern along the filament z-axis suggests extensibility and assembly variability." And "the amorphous matrix was observed only in close proximity to ECM fibers... This suggests a co-assembly between the components in these entities." These are unnecessary overinterpretations of the data. I recommend that the authors rephrase, especially when the structures cannot be identified.

Multiple studies have suggested different ECM fibers to be extensible and variable in their repeat pattern and diameter, especially when responding to forces applied by cells. We would therefore like to keep this sentence in the manuscript, but have adapted it and added further references to underline this point:

*“The variation in both diameter and repeat pattern along the filament z-axis suggests extensibility and assembly variability, **matching several studies on the properties of both fibronectin and fibrillin (Glab and Wess, 2008; Klotzsch et al., 2009; Sherratt et al., 2001; Dzamba and Peters, 1991).**”*

Similarly, we still would like to mention the potential of a co-assembly of molecules in the amorphous matrix and ECM fibers, and have rephrased the sentence accordingly to tone it down: **“Based on these observations, it is tempting to speculate that the components in these entities might co-assemble.”**

Page 6, sections "The absence of other filament-like structures in CDMs": here it would be useful to explicitly spell out the candidates from proteomics measurements, or the previous reports the authors mention (without providing references!) and the expected structural features, potentially even from structure predictions. For example, Table 2 "ECM proteins in TIFF CDMs identified by MS" could perhaps be presented in a plot form that also reflects relative abundance (possibly only the fibril-forming proteins, while others are aggregated, for simplicity) in the main text and related to the related volume fractions occupied by the different ECM structures as observed by cryo-ET.

We thank the reviewer for this comment. The citations had initially been stated in Table S1 alone and have now been added also to the main text for clarification, with a reference to Table S1 for more details:

*“Most strikingly, our data does not contain any other clearly discernible ECM fiber types, as one would expect based on our proteomics data or previous reports of the filamentous assemblies present within the ECM (Baldock et al., 2003; Lansky et al., 2019a; Sherratt et al., 2001; Früh et al., 2015) (such as FN fibers or Col-VI, **see also Table S1 for published reports on filament spacing and periodicity).**”*

Structure predictions of fibronectin, Col VI, and fibrillin are unfortunately not a viable option to define the exact structural conformation of these proteins. For example, the AlphaFold prediction of Fibronectin does not show a fibril arrangement, that could be easily compared to filaments observed in our tomograms. (<https://alphafold.ebi.ac.uk/entry/P02751>).

Moreover, these fibers are supposed to be flexible, adding another layer of difficulty to the predictions, as one would not know which state of the protein will be predicted, and which one we might actually see in our samples. Given these reasons, we did not include any AlphaFold predictions in our manuscript.

For example, Table 2 "ECM proteins in TIFF CDMs identified by MS" could perhaps be presented in a plot form that also reflects relative abundance (possibly only the fibril-forming proteins, while others are aggregated, for simplicity) in the main text and related to the related volume fractions occupied by the different ECM structures as observed by cryo-ET.

The suggestion to plot the MS data reflecting the relative MS abundance and to relate this to the volume fractions of the different entities in our tomograms is very interesting. While, this indeed could help to get a better understanding of how much the different components contribute to form structures observed in our data, there are several limitations associated with this approach, which we discuss below.

We have generated plots based on our proteomics data showing the relative abundance of fibril-forming ECM proteins with respect to each other, and also with respect to all (fibril-forming and non-fibril-forming) ECM proteins. We note, that our proteomics data allows to determine relative abundance, and with this in mind we created a semi-quantitative comparison between the different CDM components. We have added this comparison as two panels in Figure 1:

Figure 1: On-grid CDM generation and characterization.

A) Schematic depiction of CDM growth on EM grids (see Materials and Methods for details). **B)** Time-course of on-grid collagen fiber deposition over 19 days. CDMs were live-stained with the collagen binding protein CNA35-EGFP and imaged by confocal microscopy. Shown here are maximum intensity Z-projections of exemplary CDM areas on different representative EM grids on Day 0 (start of ascorbic acid treatment after reaching cell confluency), Day 7, Day 14, and Day 19. **C)** TIFF CDMs were grown for 14 days, fixed, and stained with an anti-FN I antibody, and DAPI and phalloidin to visualize the nucleus and actin cytoskeleton, respectively. Specimens were imaged by confocal microscopy and an exemplary region of a CDM is shown here as maximum intensity Z-projection of each staining as well as a merge of all three stainings. **D, E) Semi-quantitative comparison between different ECM components in Day 14 TIFF CDMs as determined by mass spectrometry. D) A pie-chart comparison of the relative amounts of fibrillar ECM proteins versus other ECM proteins in CDMs. E) A pie-chart comparison of the relative abundance of the different fibrillar ECM proteins found in our CDMs.** Scale bar dimensions are shown in the figure.

We now include these plots as new panels D and E in an updated version of Figure 1, as they indeed allow a straightforward interpretation of ECM content.

However, concerning the quantification of volume fractions in our cryo-electron tomograms, we need to point out that the tomograms presented in this study were acquired at selected sites, based on the quality (such as thickness) of the lamella and what we defined as sites of interest when visualizing lamellae at lower magnification. This means that our selection of acquisition sites was influenced by the visibility of features of interest, such as fibrillary ECM structures, or intriguing sites of cell-ECM interaction.

Hence, the tomograms do not allow a generalization of the distribution of the individual ECM components, but instead are aimed at describing their molecular architecture and interplay. Thus, a comparison between tomogram volume fractions and MS volume fractions could be potentially misleading. We have chosen to include a quantification comparing volume fractions in our tomograms for the reviewer (as described below), but would not include them in the manuscript:

Figure R1: 10 exemplary tomograms were chosen from the 42 available tomograms, 5 of which had mostly a cross-section view of ECM fibers and 5 a transverse view. For each tomogram, 5 central slices were manually measured for the area occupied by collagen, small ECM fibers, amorphous density, and seemingly empty area in the ECM. The calculated fractions per tomogram were averaged. Then, an average of all 10 tomograms was calculated and is displayed as a pie chart.

Importantly, with the workflow now being established, such a quantitative analysis as suggested by the reviewer is definitely something that could be pursued in future work, keeping the limitations of comparing volume data (of proteins with different density) to proteomics quantifications in mind.

Indeed, in our new data acquisitions we already take this into consideration (e.g. by performing unbiased montage cryo-ET to image entire lamellae).

We now mention in the Discussion how montage cryo-ET can increase throughput, together with the mentioning of the Waffle-method and Serial-liftout (please see answer below).

I recommend that the authors include in their discussion/conclusion some mention on the potential of "waffle" or "Serial lift-out" methodologies to gain higher throughput and more insights into similar model systems given the dimensions of their specimens.

We now discuss both methods in the revised manuscript.

“A combination of the workflow shown in this study and recently introduced methodologies such as the “Waffle method” (Kelley et al., 2022), the “Serial lift-out” technique (Schjøtz et al., 2023) and montage cryo-electron tomography (Peck et al., 2022; Yang et al., 2023; Chua et al., 2024) could substantially improve throughput. In turn, with increased dataset sizes this could grant more detailed and potentially also quantitative insights into CDMs and their components.”

Movies S4 and S4: some of the fuzzy amorphous density appears to be continuous with the Col-I fibrils. It might be useful for the authors to mention this if this appears to be a common feature in other data as it might indicate that these structures are precursors of Col-I assembly.

The amorphous density that appears to be following/preceding the Col-I fibrils can be observed in several instances, often when the fibrils are running in transverse. However, this is not the case for all fibrils. This makes the amorphous density closely associated with some Col-I fibrils difficult to interpret. It could show potential precursor structures, but it could also show bound sugars and other proteins. Hence, we refrain from speculating about it.

Figure S1: scale bar size missing in b.

Corrected.

Methods, cryo-FIB: unclear at which point the half-moon grid are mounted into autogrids. Please add.

A sentence was added to clarify.

“Prior to sample loading, a half-moon grid with 4 finger-like extensions for lift-out attachment (Ted Pella, #10GC04) was clipped into an AutoGrid. For clipping, all finger-like extensions of the

half-moon grid were oriented in line with the milling window to allow for low angle sample thinning.”

Data availability: it would be extremely valuable for the community if the authors deposit the tomograms and segmentations associated with this work. I would even strongly recommend deposition of the full data, from raw movies to segmentations in EMPIAR.

We have deposited the raw frames, tilt series and isonet-corrected tomograms to EMPIAR. They will be available via accession code EMPIAR-11897 upon publication of this manuscript. We hope that this data will represent a valuable resource to the scientific community interested in ECM and cryo-electron tomography.

Reviewer #2 (Comments to the Authors (Required)):

In this article entitled "Unveiling the ultrastructural landscape of extracellular matrix via lift-out cryo-FIBSEM and cryo-ET", Zens et al. reporting a technical development based on cryoEM and FIBSEM technologies. They adapted and combined methodologies (references 24, 28, 31 and 36) to enable visualization of cell derived matrices in a hydrated context. This is of outmost relevance considering the deleterious effects of classic protocols including chemical fixation and dehydration steps.

This article is well written and well-illustrated. It has an appreciable part of "trial and error" reporting of the scientific process, and results in establishment of a successful and promising protocol for the study and for future studies. This is the primary merit of the study and as such may become seminal in the field.

Limitations of the manuscript are related to the relatively low number of conclusive points that are made by the authors. Indeed, the use of a single biological condition (TIFF derived CDM) is somewhat limiting the general value of the study. Having said that, owing to the laborious nature of this approach, limiting efforts to a single sample time is understandable. Major comments:

The authors should consider toning down this statement:

"Another aggravating factor for the structural annotation of ECMs is the heterogeneity of tissue-derived ECM material. In contrast, cell derived matrices (CDMs) are a highly adaptable and versatile tool that is increasingly used to recapitulate the complexity of native tissue ECM (29-31)."

In reality, CDM are not recapitulating the complexity of native tissue, which is the results of many cell types interacting together and jointly contributing to the matrix compositions and remodelling. Only in vivo observation can fully recapitulate in vivo complexity and CDM is rather a (valuable) reductionist in vitro model system to study matrix biology.

The statement was rephrased and toned down as follows:

*"Another aggravating factor for the structural annotation of ECMs is the heterogeneity of tissue-derived ECM material. In contrast, cell derived matrices (CDMs) are a highly adaptable and versatile tool **that is increasingly used to mimic fundamental aspects of native tissue ECM and study its structure and function (Hakkinen et al., 2011; Petrie and Yamada, 2016; Kaukonen et al., 2017; Cukierman et al., 2001).**"*

The authors are taking care not to overinterpret their findings and this is commendable. However, it would be beneficial for the reader to if some more concrete conclusions could be drawn. some conclusions out of this work.

Related to this topic, the "beaded filaments" and the absence of fibronectin visualization are surprising and need more explanation.

Can the authors rule out the possibility that "amorphous ECM" is in fact not well-preserved during sample preparation? Can the rare fibers that are observed in the amorphous ECM be a low percentage of preserved material?

To our knowledge, our work is the first study to employ cryo-electron tomography on natively preserved (i.e. fully vitrified) ECM at this resolution. There are no other studies we can directly compare our data to. Also, there are currently no high-resolution structures available for fibronectin fibers, fibrillin fibers, or collagen VI fibers that we could use as gold-standard for comparison. As explained above, AlphaFold predictions are also not providing unambiguous structures for comparison.

While we aimed to avoid any damage to our specimen and to keep it as natively preserved as possible, we naturally cannot exclude (as is the case for any experimental approach) that specimen preparation induces minor alterations. However, we are confident that the workflow we have established is as optimal as currently possible.

Specifically, we do not fix or otherwise chemically treat our specimen during preparation. Also, we have tested different buffer conditions for cryo-protection, including one that contained no additional cryo-protectant but solely cell culture medium (which showed very good contrast, but as noted good but still incomplete vitrification). In all these conditions the amorphous ECM is present.

The CNA35-EGFP live stain has been used in a number of studies for the visualization of collagen fibers (e.g. Raghuraman et al., 2022, PMID: 34995086; Conway et al., 2023, PMID: 37436978; Peuhu et al., 2022, PMID: 36283390; Staneva et al., 2018, PMID: 30303750; Hsu et al., 2022, PMID: 364000786) and no detrimental effects have been reported.

In summary, all of the above leads us to the belief that the amorphous density is not an artefact created by our sample preparation process.

Can this be fibronectin filaments?

Can the beaded filaments be fibronectin filaments?

We would like to answer both these questions together.

It is possible that the rare filaments in the amorphous ECM are fibronectin filaments and this has been a thought we have discussed multiple times. However, due to the lack of structural knowledge of fibronectin we cannot make any definite statements.

Similarly, we also cannot exclude the possibility that the beaded filaments are indeed fibronectin. However, as stated in the manuscript the periodicity we observed fits better to fibrillin fibers.

There are different experimental approaches that we could take in the future to further identify these filaments. This could for example include immunogold labeling approaches. We did indeed try this with an anti-Fibronectin antibody, but unfortunately without achieving sufficient labelling efficiency and specificity to make an unambiguous statement. Future work could invest into optimizing this approach, requiring substantial experimental efforts, which we believe is beyond the scope of this work. Specifically, based on our experience with this approach we are not

convinced that immunogold labeling would be a straightforward method for identifying single structures in highly complex, tightly packed environments such as our CDM.

Another approach could be to employ more advanced super-resolution cryo-fluorescence microscopy methods once they become available (as we would require resolution in the sub-20 nanometer range in 3D to achieve unambiguous identification).

Finally, as already pointed out by the reviewer in the next comment, genetic engineering of CDMs via knockout of specific proteins could be used as an indirect experimental approach to identify filaments (e.g. via the absence of specific features upon KO of Fibronectin, Fibrillin etc.). As stated below, this is indeed an avenue we are following up, but is beyond the scope of this manuscript.

One major limitation of the study is the lack of comparisons between for example condition or treatments that are expected to influence the CDM. For example, genetic manipulation of the TIFF or CDM made by another cell type. If these are beyond the scope of this study, the authors should discuss this point and mention this limitation.

We fully agree that the genetic manipulation of TIFF cells for the CDM production or the use of another cell type would be an important future step for this project. The time-consuming and complex process for sample generation and data acquisition render these options being out of scope for this manuscript (as also acknowledged by this reviewer).

Indeed, we had already tried to address these future options in the use of CDMs in the main text. We have now added that these plans are beyond the scope of the current manuscript (**changes in bold**):

“As also revealed by our data, ECM filaments represent a challenging target for structure determination approaches due to their apparent variability. Future experiments using lift-out cryo-FIBSEM and cryo-ET, combined with novel image processing tools based on neural networks (for example (Rice et al., 2023)) or functional studies using genetic knockouts will be required to extend and annotate the gallery of ECM structures.”

*“Specifically, based on the hypothesis that changes in matrix composition of ECMs of different origin are reflected on a structural level, cryo-ET of CDMs now allows studying how ECM-specific topologies define tissue homeostasis and underlying cellular behavior. Hence, CDMs derived from cell types of different origin, e.g. skin, lung, mammary or cancer-associated fibroblasts (CAF), offer an appealing avenue for comparative analysis of ECMs to obtain a clearer depiction of the role of individual ECM components in defining specialized tissue-specific matrices. Specifically, this could be achieved via an integrative approach using a combination of molecular imaging via cryo-ET, proteomics analysis via mass spectrometry (MS) followed by functionally characterizing the role of specific matrix components using CRISPR-Cas9 knockout approaches. Together with pharmacological treatment to target specific ECM component this offers the possibility to manipulate ECM production. **While being beyond the scope of this work, these approaches offer exciting future avenues for a more detailed understanding of the ECM.**”*

Minor comments:

The authors abstract first paragraph is somewhat misleading, especially when mentioning Col VI assembly as remaining "enigmatic", considering that there is no conclusive statement made in the manuscript regarding Col VI.

We have removed this statement.

Fig S1 : Please include original images of panel A, before annotation.

Fig S1 was updated accordingly.

“Figure S1: Room-temperature SEM array tomography of TIFF CDMs grown for 14 days.

A) Four exemplary 70 nm thick sections from different z-height positions within the CDM are shown. B) The same slices shown in (A) were segmented for nuclei, cytoplasm, and ECM using Ilastik (Berg et al., 2019) and are colored as indicated. C) 3D segmentation of an exemplary cell embedded within the CDM (denoted by a black asterisk in A), shown in a top (left) and oblique view (middle). On the right side, an ortho-slice view of the same cell is shown, highlighting the thin z-height dimension of the cell embedded within the ECM. Please note that the other cells in vicinity of the segmented cell have been omitted from this view to facilitate visualization. Color code of the

segmentation is indicated below. D) The directionality of cells and fibers is color-coded as indicated on the right, corresponding to the angles of cells and fibers. The same z-positions as shown in (A) are depicted to show the change in directionality throughout the height of the CDM. Scale bar dimensions are annotated in the figure."

Fig S6 : "enodcytosis" should be corrected to "endocytosis".

Corrected.

Fig S9 : If available, please show low magnification pictures for panel D. The same way it has been done in Fig S6 A, for instance.

Fig S9/panel D is a full tomogram and not a higher magnification image of a tomogram. Due to this, we cannot include a low magnification image similar to Fig S6/A.

These expressions/sentences are unclear or inappropriate to my opinion.
"There is no unambiguous consensus..."
"molecular sociology..."

The sentences were rephrased to avoid the mentioned expressions:

"current literature does not allow for an unambiguous consensus on the exact molecular assembly of many of the other ECM fibers"

"This includes the structural and functional characterization of **single components and their interactions** in natively preserved ECM."

Reviewer #3 (Comments to the Authors (Required)):

This study is an exquisite example of how new technology can facilitate novel information. The only areas of potential improvement are some missing citations to credit the original CDM work and methodology (and similar) as well as a suggestion to combine the Cryo-generated data with SHG (second harmonic generation of polarized light) using the same 3D sample. The latter is merely a suggestion that could significantly improve the interpretation of the data (regarding collagen fibers).

We appreciate the reviewer's positive comments.

Specific MINOR suggestions:

1. In the introduction, when CDMs are first described as a valid in vivo mimicry approach, the authors may want to cite the original paper and methods depicting these cell/ECM functional units:

a. Paper: doi: 10.1126/science.1064829

b. Methods:

i. Original 2002: <https://doi.org/10.1002/0471143030.cb1009s16>

ii. Updated in 2007: <https://doi.org/10.1002/0471143030.cb1009s33>

iii. Updated again in 2016: PMID: PMC5058441

We thank the reviewer for this input and have added the required citations.

2. A few sentences later in the introduction, when highlighting some "fundamental research" examples, the authors should include cancer research and cite some of the Cukierman team's work:

a. *elife* 2017

b. *Matrix Biology* 2019

c. *Communications Biology* 2020

d. *Cancer Discovery* 2021

e. *CRC* 2022

We apologize for not having cited these papers already in the first version of our manuscript. We have now adapted the text to include these citations and mention cancer research in this context: *"The high versatility and adaptability of CDMs to distinct research questions has also made them a routinely used tool in cancer research (Franco-Barraza et al., 2017; Malik et al., 2019; Padhi et al., 2020; Francescone et al., 2021; Jones et al., 2022). The use of cancer-associated fibroblasts (CAFs) for CDM generation results in a close mimic of the tumor-associated stroma (Amatangelo*

et al., 2005) and has aided in the identification of novel regulators of ECM alignment (Jones et al., 2022) as well as potential therapy targets (Francescone et al., 2021)."

3. Technically, in results shown in Figure 1C, the day 14 fibronectin staining looks as if the immunofluorescence followed a process whereby fixing of the cells was done first and permeabilization followed. The original methods have provided troubleshooting for this process (to avoid the apparent void in fibers) by conducting a simultaneous fix/permeabilization step followed by added fixing (published in this version of CDM production methods and quality control analyses: PMID: PMC5058441). The authors may want to repeat this using this advice and potentially generate a higher quality image/figure.

We thank the reviewer for pointing us towards these papers and the protocol for a better way of fixation/permeabilization. We have repeated the day 14 staining of Figure 1C accordingly. This indeed substantially improved the image quality. We have updated Figure 1, Panel C and describe this also in the methods. Figure 1 has been included in the answer to Reviewer #1, including Panel C that was updated (see above).

4. Regarding data in supplemental figure 2, CDMs have been shown to mature enough (e.g., incorporate enough fibrous collagen) in vitro to include collagen fibers that are detected using the second harmonic generation of polarized light (obtained with most multiphoton microscopes: PMID: PMC7442735). Hence, the authors may want to include this orthogonal approach to support their interpretation of the data (to support that the 25-60 nm filaments are indeed collagen fibers as suggested) and demonstrate the high quality of their CDMs.

As suggested, we have performed second harmonic generation of our CDMs to visualize collagen fibers. Specifically, we have combined our CNA35-EGFP live staining of collagen with second harmonic generation of polarized light.

As expected, the SHG imaging confirms the results obtained by using the collagen live-stain. The collagen shown by SHG imaging overlaps with our fluorescent signal for CNA35-EGFP, highlighting that both imaging methods visualize the same fibers.

Considering the fluorescence microscopy data and SHG imaging data of our CDMs, combined with the clearly visible D-spacing in the collagen fibers in our cryo-electron tomograms, we are confident that the filaments we visualize are indeed collagen fibers.

The main advantage of the CNA35-EGFP imaging is that it is compatible with our cryo-fluorescence approach that we use prior to ion-beam milling and lift-out experiments, and that it requires substantially less laser power which is beneficial for specimen preservation. Given that SHG and fluorescent imaging achieve approximately the same resolution and due to space constraints with showing supplementary data, we have refrained from adding the SHG imaging to the manuscript.

Figure R2: On-grid imaging of collagen in CDMs.

TIFF CDMs were grown for 14 days and stained with CNA35-EGFP as described in the manuscript. Specimens were then imaged on a Leica SP8 DIVE Multiphoton and confocal microscope, using a HC FUOTAR L 25x / 0.95 W objective. Two exemplary sites of two different CDMs are shown here, imaged first by second-harmonic generation of polarized light microscopy (SHG) and then by fluorescence microscopy. Scale bar dimensions are shown in the figure.

5. The statement suggesting that the growth of new CDM (fibers) preferentially takes place on the top layers is not well justified and should probably be omitted, while the density of fibers statement could prevail.

As suggested by the reviewer, we have now rephrased this section and removed the mentioning of CDM growth in the top layers. Instead, we just mention an apparent decrease in fiber density towards the top. Please also see our response to reviewer #1.

“Empty areas in extracellular space devoid of any structures were also observed, and a decrease in ECM fibre density from bottom towards top of the CDM could be observed in a majority of lamellae.”

February 23, 2024

RE: JCB Manuscript #202309125R

Dr. Florian KM Schur
Institute of Science and Technology Austria
Am Campus 1
Klosterneuburg 3400
Austria

Dear Dr. Schur,

Thank you for submitting your revised manuscript entitled "Unveiling the ultrastructural landscape of extracellular matrix via lift-out cryo-FIBSEM and cryo-ET". We would be happy to publish your paper in JCB pending final revisions necessary to meet our formatting guidelines (see details below).

A. MANUSCRIPT ORGANIZATION AND FORMATTING:

1) Text limits: Character count for Reports is < 20,000, not including spaces. Count includes title page, abstract, introduction, combined results & discussion, and acknowledgments. Count does not include materials and methods, figure legends, references, tables, or supplemental legends.

2) Figure formatting: Reports may have up to 5 main text figures. Scale bars must be present on all microscopy images, including inset magnifications. Please add a scale bar for the magnification image in Figure S1C.

Also, please avoid pairing red and green for images and graphs to ensure legibility for color-blind readers. If red and green are paired for images, please ensure that the particular red and green hues used in micrographs are distinctive with any of the colorblind types. If not, please modify colors accordingly or provide separate images of the individual channels.

3) Statistical analysis: Error bars on graphic representations of numerical data must be clearly described in the figure legend. The number of independent data points (n) represented in a graph must be indicated in the legend. Please, indicate whether 'n' refers to technical or biological replicates (i.e. number of analyzed cells, samples or animals, number of independent experiments). If independent experiments with multiple biological replicates have been performed, we recommend using distribution-reproducibility SuperPlots (please see Lord et al., JCB 2020) to better display the distribution of the entire dataset, and report statistics (such as means, error bars, and P values) that address the reproducibility of the findings.

Statistical methods should be explained in full in the materials and methods. For figures presenting pooled data the statistical measure should be defined in the figure legends. Please also be sure to indicate the statistical tests used in each of your experiments (both in the figure legend itself and in a separate methods section) as well as the parameters of the test (for example, if you ran a t-test, please indicate if it was one- or two-sided, etc.). Also, if you used parametric tests, please indicate if the data distribution was tested for normality (and if so, how). If not, you must state something to the effect that "Data distribution was assumed to be normal but this was not formally tested."

4) Title: To make the title more concise and match our preferred style we suggest a slight change to the following: "Lift-out cryo-FIBSEM and cryo-ET reveal the ultrastructural landscape of extracellular matrix"

5) Materials and methods: Should be comprehensive and not simply reference a previous publication for details on how an experiment was performed. Please provide full descriptions (at least in brief) in the text for readers who may not have access to referenced manuscripts. The text should not refer to methods "...as previously described."

6) For all cell lines, vectors, constructs/cDNAs, etc. - all genetic material: please include database / vendor ID (e.g., Addgene, ATCC, etc.) or if unavailable, please briefly describe their basic genetic features, even if described in other published work or gifted to you by other investigators (and provide references where appropriate). Please be sure to provide the sequences for all of your oligos: primers, si/shRNA, RNAi, gRNAs, etc. in the materials and methods. You must also indicate in the methods the source, species, and catalog numbers/vendor identifiers (where appropriate) for all of your antibodies, including secondary. If antibodies are not commercial, please add a reference citation if possible.

- 7) Microscope image acquisition: The following information must be provided about the acquisition and processing of images:
- Make and model of microscope
 - Type, magnification, and numerical aperture of the objective lenses
 - Temperature
 - Imaging medium
 - Fluorochromes
 - Camera make and model
 - Acquisition software
 - Any software used for image processing subsequent to data acquisition. Please include details and types of operations involved (e.g., type of deconvolution, 3D reconstitutions, surface or volume rendering, gamma adjustments, etc.).
- 8) References: There is no limit to the number of references cited in a manuscript. References should be cited parenthetically in the text by author and year of publication. Abbreviate the names of journals according to PubMed.
- 9) Supplemental materials: Reports generally have up to 3 supplemental figures and 10 videos. You currently exceed this limit but, in this case, we will be able to give you the extra space. Please also note that tables, like figures, should be provided as individual, editable files. A summary of all supplemental material should appear at the end of the Materials and methods section. Please include one brief sentence per item.
- 10) Video legends: Should describe what is being shown, the cell type or tissue being viewed (including relevant cell treatments, concentration and duration, or transfection), the imaging method (e.g., time-lapse epifluorescence microscopy), what each color represents, how often frames were collected, the frames/second display rate, and the number of any figure that has related video stills or images.
- 11) eTOC summary: A ~40-50 word summary that describes the context and significance of the findings for a general readership should be included on the title page. The statement should be written in the present tense and refer to the work in the third person. It should begin with "First author name(s) et al..." to match our preferred style.
- 12) Conflict of interest statement: JCB requires inclusion of a statement in the acknowledgements regarding competing financial interests. If no competing financial interests exist, please include the following statement: "The authors declare no competing financial interests." If competing interests are declared, please follow your statement of these competing interests with the following statement: "The authors declare no further competing financial interests."
- 13) A separate author contribution section is required following the Acknowledgments in all research manuscripts. All authors should be mentioned and designated by their first and middle initials and full surnames. We encourage use of the CRediT nomenclature (<https://casrai.org/credit/>).
- 14) ORCID IDs: ORCID IDs are unique identifiers allowing researchers to create a record of their various scholarly contributions in a single place. Please note that ORCID IDs are required for all authors. At resubmission of your final files, please be sure to provide your ORCID ID and those of all co-authors.
- 15) Journal of Cell Biology now requires a data availability statement for all research article submissions. These statements will be published in the article directly above the Acknowledgments. The statement should address all data underlying the research presented in the manuscript. Please visit the JCB instructions for authors for guidelines and examples of statements at (<https://rupress.org/jcb/pages/editorial-policies#data-availability-statement>).

B. FINAL FILES:

****The license to publish form must be signed before your manuscript can be sent to production. A link to the electronic license to publish form will be sent to the corresponding author only. Please take a moment to check your funder requirements before choosing the appropriate license.****

Thank you for this interesting contribution, we look forward to publishing your paper in Journal of Cell Biology.

Sincerely,

Kenneth Yamada, MD, PhD
Editor
Journal of Cell Biology

Dan Simon, PhD
Scientific Editor
Journal of Cell Biology